# Roles of the ClC chloride channel CLH-1 in food-associated salt chemotaxis behavior of *C. elegans*

Chanhyun Park[1†], Yuki Sakurai[1], Hirofumi Sato[1], Shinji Kanda[1,2], Yuichi Iino[1*], Hirofumi Kunitomo[1*]

[1]Department of Biological Sciences, School of Science, The University of Tokyo, Tokyo, Japan; [2]Laboratory of Physiology, Atmosphere and Ocean Research Institute, The University of Tokyo, Chiba, Japan

**Abstract** The ability of animals to process dynamic sensory information facilitates foraging in an ever-changing environment. However, molecular and neural mechanisms underlying such ability remain elusive. The ClC anion channels/transporters play a pivotal role in cellular ion homeostasis across all phyla. Here, we find a ClC chloride channel is involved in salt concentration chemotaxis of *Caenorhabditis elegans*. Genetic screening identified two altered-function mutations of *clh-1* that disrupt experience-dependent salt chemotaxis. Using genetically encoded fluorescent sensors, we demonstrate that CLH-1 contributes to regulation of intracellular anion and calcium dynamics of salt-sensing neuron, ASER. The mutant CLH-1 reduced responsiveness of ASER to salt stimuli in terms of both temporal resolution and intensity, which disrupted navigation strategies for approaching preferred salt concentrations. Furthermore, other ClC genes appeared to act redundantly in salt chemotaxis. These findings provide insights into the regulatory mechanism of neuronal responsivity by ClCs that contribute to modulation of navigation behavior.

*For correspondence:
kunitomo@bs.s.u-tokyo.ac.jp

Present address: †Center for Nanomedicine, Institute for Basic Science (IBS), Seoul, Republic of Korea

## Introduction

Generating an optimal foraging behavior based on experiences is basic and important ability for survival. Mechanisms of food-associated learning have long been addressed in many species, dating back to Pavlovian appetitive conditioning demonstrated in dogs (*Braubach et al., 2009*; *Cho et al., 2016*; *Gottfried et al., 2003*; *Hirano et al., 2013*; *O'Doherty et al., 2003*; *Otis et al., 2017*; *Pavlov, 1927*; *Sasakura and Mori, 2013*; *Winter and Stich, 2005*). By virtue of its simple nervous system and amenability to genetic manipulations, the soil nematode *Caenorhabditis elegans* has been used to unveil molecular and neural mechanisms of behavior. *C. elegans* shows food-associated behavioral plasticity in combination with various sensory modalities including gustatory, olfactory, thermosensory, and mechanosensory cues (*Colbert and Bargmann, 1995*; *Hedgecock and Russell, 1975*; *Kindt et al., 2007*; *Saeki et al., 2001*). We have previously reported that *C. elegans* shows plasticity in chemotaxis to salt (sodium chloride; NaCl); wild-type animals are attracted to the salt concentration at which they have been fed, while avoid the concentrations at which they have been starved (salt concentration chemotaxis, *Kunitomo et al., 2013*, *Figure 1a*). *C. elegans* senses inorganic ions mainly through the bilateral salt-sensing neuron pair, ASE (*Bargmann and Horvitz, 1991*). Sensory input to ASE-right (ASER) is essential and sufficient for food-associated salt chemotaxis. The interneurons postsynaptic to ASER, namely, AIA, AIB, and AIY, regulates exploratory behaviors (*Gray et al., 2005*; *Li et al., 2014*; *Piggott et al., 2011*). Modulation of synaptic transmission between ASER and these interneurons is involved in modification of salt chemotaxis (*Kunitomo et al., 2013*; *Luo et al., 2014*; *Wang et al., 2017*).

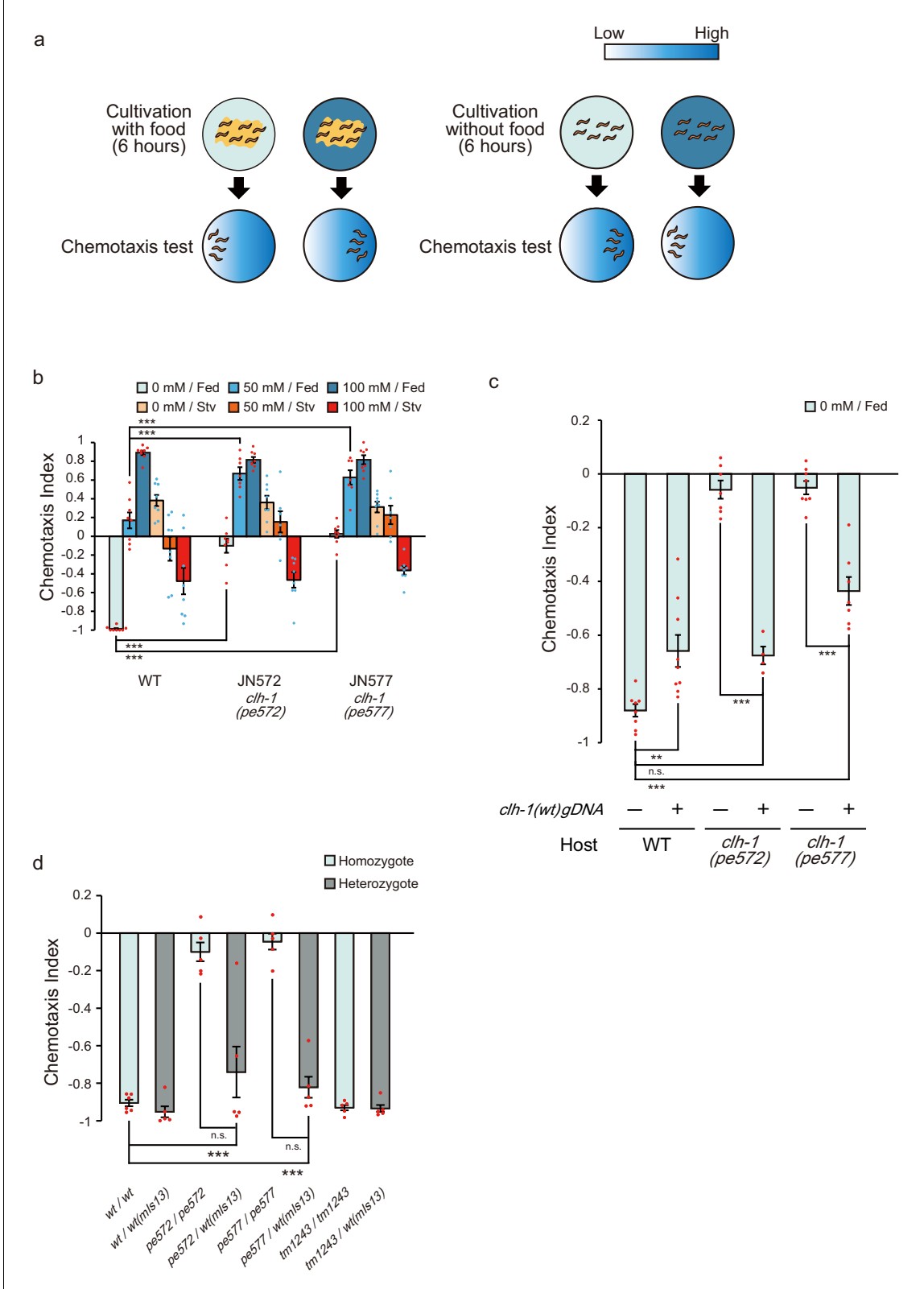

**Figure 1.** Two missense mutations in *clh-1* give rise to food-associated salt chemotaxis disorder. (**a**) Salt concentration chemotaxis of wild type. Adult hermaphrodites were cultivated at 0 mM or 100 mM of NaCl with or without food for 6 hr and placed on a chemotaxis assay plate on which NaCl gradient was created. Distribution of animals was quantified by calculating a chemotaxis index. See Materials and methods for details. (**b**) Chemotaxis of wild-type animals and two mutants obtained from screening, JN572: *clh-1(pe572)* and JN577: *clh-1(pe577)*. Dots represent individual trials. Bars and

*Figure 1 continued*

the error bars represent mean +/- s.e.m., *n* = 8 assays, Dunnett's test, ***p<0.001. (c) Rescue of *clh-1(pe572)* and *clh-1(pe577)* mutants by a *clh-1* genomic DNA fragment. Dots represent individual trials. Bars and the error bars represent mean +/- s.e.m., *n* ≥ 4, Tukey's test, ***p<0.001, **p<0.01, n.s. not significant. (d) Chemotaxis of *clh-1* heterozygotes. *clh-1* homozygotes were crossed with a *clh-1(wt)* reporter strain that express GFP in pharyngeal muscle (*mIs13*). Resulted F1 animals were used for assay. Dots represent individual trials. Bars and the error bars represent mean +/- s.e.m., *n* ≥ 4, Tukey's test, ***p<0.001, n.s. not significant.

The online version of this article includes the following figure supplement(s) for figure 1:

**Figure supplement 1.** Isolation and characterization of salt chemotaxis mutants JN572 and JN577.
**Figure supplement 2.** Missense mutations in *clh-1* responsible for salt chemotaxis defect.
**Figure supplement 3.** Characterization of *clh-1* mutants.

Molecular mechanisms for perception and propagation of salt stimuli in ASER have been proposed. A receptor-type guanylyl cyclase GCY-22 (also known as CHE-5) plays a pivotal role in perception of salt stimuli and is suggested to act as an ion receptor of ASER (*Adachi et al., 2010*; *Kunitomo and Iino, 2020*; *Ortiz et al., 2009*; *Smith et al., 2013*). Excitation of ASER depends also on cyclic nucleotide-gated (CNG) channels consisting of TAX-2 and TAX-4 (*Suzuki et al., 2008*). *C. elegans* genome does not contain typical voltage-gated sodium channel genes (*Goodman et al., 1998*). Instead, voltage-gated calcium channels are responsible for propagation of depolarization, which is examined by electrophysiological studies (*Goodman et al., 1998*; *Shindou et al., 2019*). However, contribution of anions to the regulation of ASER activity has not been discussed yet.

Anion transporters play critical roles in regulating excitability of neurons as they finely tune electrophysiological properties of membranes. However, how and which molecules contribute to responsivity of specific neurons and eventually produces behavioral output, remains rudimentary. Here, we identify the ClC chloride channel CLH-1 as a possible regulator of food-associated salt chemotaxis in *C. elegans*. ClC proteins transport univalent anions across membranes to control electrochemical potential of excitable cells and to maintain ionic milieu as well as pH of intracellular organelles (*Ahnert-Hilger and Jahn, 2011*; *Bösl et al., 2001*; *Branicky et al., 2014*). Malfunction of ClC genes result in various diseases such as myotonia, leukodystrophy, hyperaldosteronism, and epilepsy in humans (*Blanz et al., 2007*; *Charlet-B et al., 2002*; *Fernandes-Rosa et al., 2018*; *Yamamoto et al., 2015*). The causal relationship among ClC malfunction, physiological consequences, and disease manifestations are not fully understood in many cases. CLH-1 shares the highest (37%) identity with mammalian ClC-2. Functional studies using heterologous expression in *Xenopus* oocytes and mammalian cells demonstrated that both CLH-1 and ClC-2 are inwardly rectifying chloride channels (*Grant et al., 2015*; *Nehrke et al., 2000*; *Staley et al., 1996*; *Thiemann et al., 1992*). On the other hand, two recent studies showed that ClC-2 contributes to Cl$^-$ influx in neurons depending on the electrochemical potential of Cl$^-$ (*Ratté and Prescott, 2011*; *Rinke et al., 2010*). Thus, the function of CLH-1/ClC-2 in the nervous system remains elusive.

In this study, we show two novel missense mutations in *clh-1* change NaCl concentration preference of *C. elegans* only after food experience. Genetic analyses revealed that CLH-1 acts in ASER and that both quantitative and anatomical localization of CLH-1 is required for normal chemotaxis. Functional imaging of neurons indicated that mutations in *clh-1* altered responsivity of the ASER and AIB neurons, which consequently affect behavioral outputs. These results suggest that responsivity of the salt circuit is maintained by CLH-1 to generate proper navigation behavior in salt chemotaxis.

## Results

### Missense mutations of *clh-1* give rise to a disorder in food-associated salt concentration chemotaxis

*C. elegans* adults cultivated at a particular NaCl concentration for 6 hr are attracted to the salt concentration if they have been fed, while avoid the concentration if they have been starved (*Figure 1a* and *Figure 1—figure supplement 1a*). To better understand molecular mechanisms of salt chemotaxis plasticity, we screened for mutants that showed defects in salt chemotaxis after feeding but not after starvation (see Materials and methods, *Figure 1—figure supplement 1b*). Two mutants, JN572 and JN577, showed a similar phenotype: an unbiased salt preference after feeding on NaCl-

free nematode growth medium (hereinafter referred to as cultivation at 0 mM NaCl). Chemotaxis to high salt after feeding on NGM with 100 mM NaCl (hereinafter referred to as cultivation at 100 mM NaCl), and chemotaxis after starvation were comparable to those of wild-type animals (*Figure 1b* and *Figure 1—figure supplement 1c*). Neither shortened nor extended cultivation at 0 mM NaCl did not ameliorate the defects (*Figure 1—figure supplement 1d,e*), suggesting that the impaired chemotaxis is not due to the delay of behavioral modification. Rather, the mutants were unable to generate migration bias toward low salt. Consistent with this idea, the mutants showed a preference for high salt after cultivation at 50 mM NaCl, under which condition wild-type animals showed an unbiased salt preference (*Figure 1b*, and see Discussion).

We mapped mutations of JN572 and JN577 between genetic positions 2.82 and 6.12 (cM) on chromosome II (see Materials and methods, *Fay and Bender, 2006*; *Wicks et al., 2001*). Genome sequencing revealed that they carried a distinct missense mutation in the *clh-1* gene, one of six ClC channels/transporters in *C. elegans*. Mutations predicted M293I and I146T substitutions in CLH-1A in JN572 and JN577, respectively, and they are hereinafter referred to as *clh-1(pe572)* and *clh-1(pe577)* (*Figure 1—figure supplement 2a,b*). Salt chemotaxis defects of the mutants were recovered by a *clh-1(wt)* genomic fragment, confirming that *clh-1* is the responsible gene, although the effect was partial in *clh-1(pe577)* (*Figure 1c*). We also noticed that extra copies of *clh-1(wt)* genomic fragment weakened low-salt chemotaxis of wild type (*Figure 1c*), implying that overexpression of CLH-1 could disrupt chemotaxis to low salt.

Interestingly, deletion mutants of *clh-1*, all of which harbor a lesion in the pore-forming transmembrane domain of CLH-1 and hence are putative loss-of-function alleles (*Figure 1—figure supplement 2a*), showed almost no discernible defect in salt chemotaxis (*Figure 1—figure supplement 3a* and *Figure 1—figure supplement 1d,e*). These results suggest that chemotaxis defects of *clh-1(pe572)* and *clh-1(pe577)* were caused by an anomalous activity of *clh-1*. To characterize the nature of *clh-1* missense alleles, we observed food-associated salt chemotaxis of heterozygotes. *clh-1(pe572)/clh-1(wt)* and *clh-1(pe577)/clh-1(wt)* showed normal low-salt chemotaxis, demonstrating that both missense alleles are recessive to wild-type allele (*Figure 1d*). Also, we noted that the effect of both missense alleles are dosage-dependent, that is, *clh-1(pe572)/clh-1(tm1243)* and *clh-1(pe577)/clh-1(tm1243)* showed a modest defect in salt chemotaxis to low salt after cultivation at 0 mM NaCl (*Figure 1—figure supplement 3b*). Furthermore, we unexpectedly found that *clh-1(tm1243)* conferred a weak resistance to an acetylcholine receptor agonist, levamisole. On the contrary, *clh-1(pe572)* or *clh-1(pe577)* caused an enhanced sensitivity to levamisole if compared with wild type, suggesting these alleles may not be simple reduction-of-function alleles (*Figure 1—figure supplement 3c*). Together, these data show that *clh-1(pe572)* and *clh-1(pe577)* (hereinafter collectively referred to as *clh-1(pe)*) are recessive mutations whose salt chemotaxis phenotype appears in a dosage-dependent manner.

Given that salt chemotaxis defects depend on the dosage of *clh-1(pe)* alleles, we wondered if overexpression of *clh-1(pe)* can give rise to salt chemotaxis defect. To examine this possibility, we introduced *clh-1(pe)* genomic DNA fragments into wild type or *clh-1(tm1243)* mutants. Overexpression of mutant *clh-1* conferred defects in low-salt chemotaxis in both genomic backgrounds, demonstrating that excess *clh-1(pe)* override the canonical *clh-1(wt)* function (*Figure 1—figure supplement 3d*). Therefore, excess *clh-1*, either wild-type or *pe* alleles, can impair low-salt chemotaxis. We concluded that *clh-1(pe572)* and *clh-1(pe577)* are atypical neomorphic alleles that disrupt chemotaxis to low salt after cultivation at 0 mM NaCl.

## Roles of *C. elegans* ClC channel/transporter genes in salt chemotaxis

All ClC anion channels/transporters characterized so far function as either homodimers or heterodimers (*Accardi, 2015*; *Stölting et al., 2014*). The *C. elegans* genome carries 6 ClC genes (*clh-1* through *clh-4* for anion channel, and *clh-5* and *clh-6* for anion transporter, predicted from a key amino acid residue and subcellular localization, *Figure 2—figure supplement 1a*, *Schriever et al., 1999*; *Nehrke et al., 2000*). This raised a possibility that mutant CLH-1 molecules impaired the function of other CLH gene products by forming heterodimers, thereby resulting in defective salt chemotaxis. To examine this, we observed salt chemotaxis of the mutants whose *clh* genes were deleted individually or in combinations with *clh-1(pe572)* or *clh-1(tm1243)* mutation. Each single mutant except for *clh-5(tm6008)* showed normal salt chemotaxis under fed conditions (*Figure 2—figure supplement 1b*). We then generated and tested a series of double (or triple in the case of *clh-3*

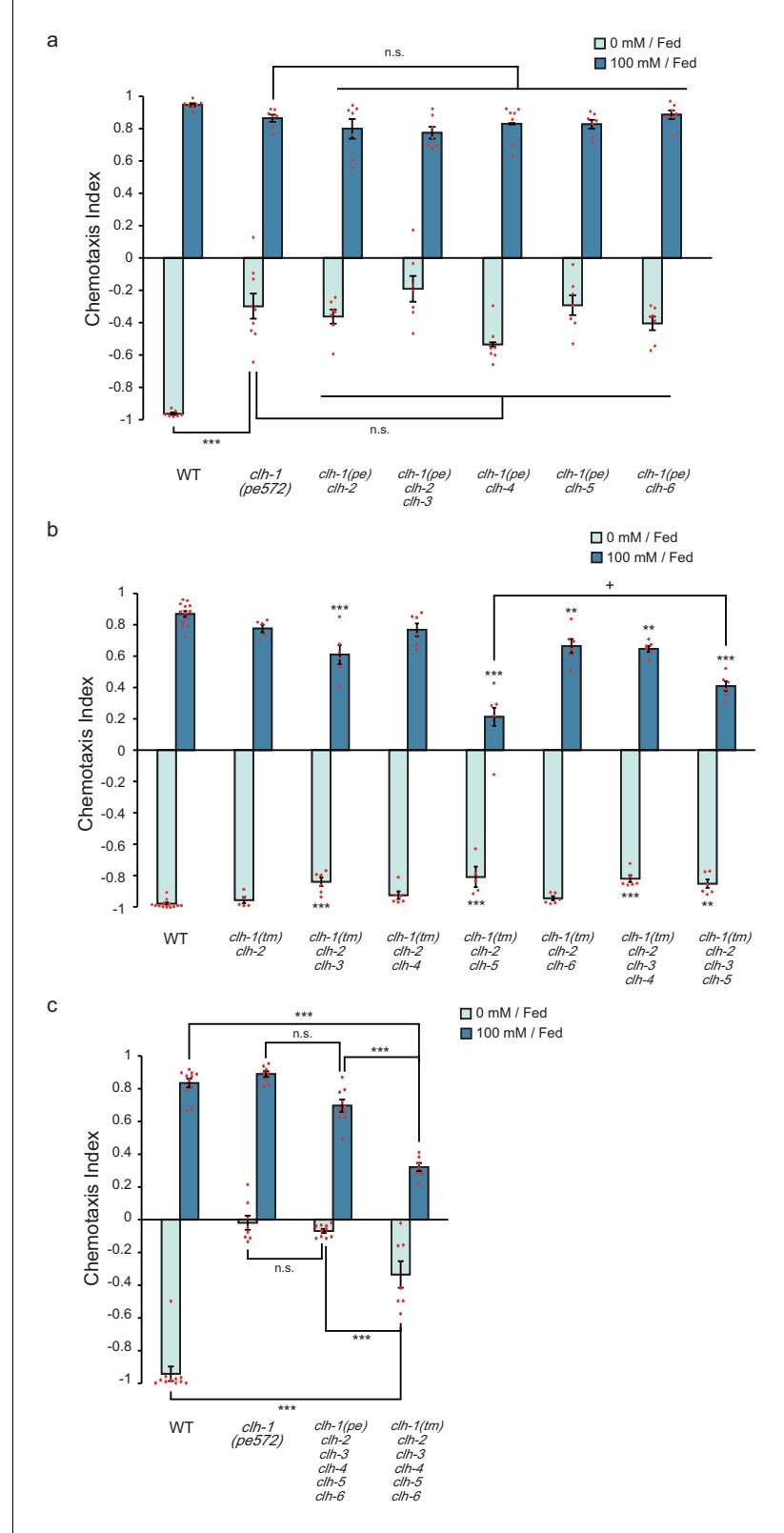

**Figure 2.** ClC genes redundantly function in salt chemotaxis. (a) Chemotaxis of *clh* multiple mutants that carry *clh-1(pe572)* mutation with a deletion in other *clh* genes. Dots represent individual trials. Bars and the error bars represent mean +/- s.e.m., *n* ≧ 6 assays, Tukey's test, ***p<0.001, n.s. not significant. (b) Chemotaxis of *clh* multiple mutants that carry *clh-1(tm1243)* mutation and a deletion in other *clh* genes. Dots represent individual

*Figure 2 continued*

trials. Bars and the error bars represent mean +/- s.e.m., $n \geqq 5$, Tukey's test, \*\*\*p<0.001, \*\*p<0.01, compared with wild type. +p<0.05, compared with indicated mutants. (c) Chemotaxis of *clh* hexatruple mutants. Dots represent individual trials. Bars and the error bars represent mean +/- s.e.m., $n \geqq 7$, Tukey's test, \*\*\*p<0.001, n.s. not significant.

The online version of this article includes the following figure supplement(s) for figure 2:

**Figure supplement 1.** Effect of loss of ClC genes on salt chemotaxis.

---

*(ok763) clh-2(ok636) clh-1(pe572))* mutants that carry *clh-1(pe572)* and a deletion in another *clh* gene (*Figure 2a*). The double mutants showed a defect similar to that of the *clh-1(pe572)* single mutant, that is, defective chemotaxis toward low salt. A hexatruple mutant that carry *clh-1(pe572)* and a deletion in five other *clh* genes also showed similar phenotype (*Figure 2c*). These results indicate that salt chemotaxis defect of *clh-1(pe572)* mutant is not attributed to impairment of other CLH proteins.

We next examined whether the *clh* genes could redundantly act in salt chemotaxis. We started from *clh-2(ok636) clh-1(tm1243)* double mutants because they had the highest homology (52% identity) among *clh* genes. Although the double mutant showed no chemotaxis defect, triple mutants with either *clh-3(ok763)* or *clh-6(tm617)* showed an impaired chemotaxis toward high salt (*Figure 2b*). Triple mutants *clh-3(ok763) clh-2(ok636) clh-1(tm1243)* and *clh-5(tm6008) clh-2(ok636) clh-1(tm1243)* showed a weak defect in chemotaxis to low salt. Interestingly, impaired chemotaxis to high salt observed in *clh-5 clh-2 clh-1* was partially restored in *clh-3(ok763) clh-5(tm6008) clh-2 (ok636) clh-1(tm1243)* quadruple mutants (*Figure 2b*). Many of these multiple mutants and *clh-5 (tm6008)* single mutants also showed a high immobility index, implying that chemotaxis might be affected by locomotion defects (*Figure 2—figure supplement 1c*, *Figure 2—figure supplement 1d*, and see Discussion). Finally, a hexatruple mutant *clh-3(ok763) clh-5(tm6008) clh-2(ok636) clh-1 (tm1243); clh-6(tm617); clh-4(ok1162)* showed severe defect in both high- and low-salt chemotaxis (*Figure 2c*). Altogether, our results suggest that *clh* genes redundantly act in salt chemotaxis. It should be noted that chemotaxis to low salt was more severely affected in *clh-1(pe572)* hexatruple mutants. This further suggests that the effect of *clh-1(pe572)* missense mutation is not simply caused by inhibition of other CLH proteins.

## CLH-1 acts in ASER to affect salt preference

It has previously been reported that *clh-1* is expressed in hypodermal cells, seam cells, D-cells of the vulva, and neuronal and glial cells of the head (*Grant et al., 2015*; *Nehrke et al., 2000*). These expression patterns were confirmed with a transcriptional reporter (*clh-1p::nls4::mTFP*, see Materials and methods). Of the head neurons, at least ASE, AWA, and AWC sensory neurons expressed the reporter (*Figure 3a*). To determine the site of action of *clh-1*, we performed cell-specific rescue experiments using *clh-1(wt)* cDNA. The mutant phenotype was rescued when cDNA was expressed either pan-neuronally or specifically in ASER, suggesting that *clh-1* acts in the nervous system including ASER (*Figure 3b*, and *Figure 3—figure supplement 1a* for ASER-specific rescue of *pe577*). On the other hand, *clh-1(wt)* cDNA failed to rescue *clh-1(pe572)* when expressed in amphid sheath (AmSh) cells, where CLH-1 function as a pH mediator (*Grant et al., 2015*), or in the left-sided ASE neuron (ASEL) (*Figure 3b*). Unexpectedly, combined expression of the transgene in ASER and AmSh cells or expression in all ciliated neurons also failed to rescue the phenotype (*Figure 3b*). These phenotypes reminded us of the impairment of low-salt chemotaxis by overexpression of *clh-1(wt)* genomic fragment (*Figure 1c*), indicating a possibility that an excess of CLH-1(wt) in the cells adjacent to ASER could disturb low-salt chemotaxis (see Discussion).

## Morphology of ASER and localization of CLH-1 are largely unaffected by missense mutations

Head sensory neurons of *C. elegans* sense environmental stimuli via neural receptive endings, which are comprised of cilia and microvilli (*Perkins et al., 1986*; *Ward et al., 1975*; *Ware et al., 1975*). Function of these endings largely depends on glia which ensheathe them. A recent study showed that AmSh glia regulate the function and shape of the AFD thermosensory neuron's microvilli by

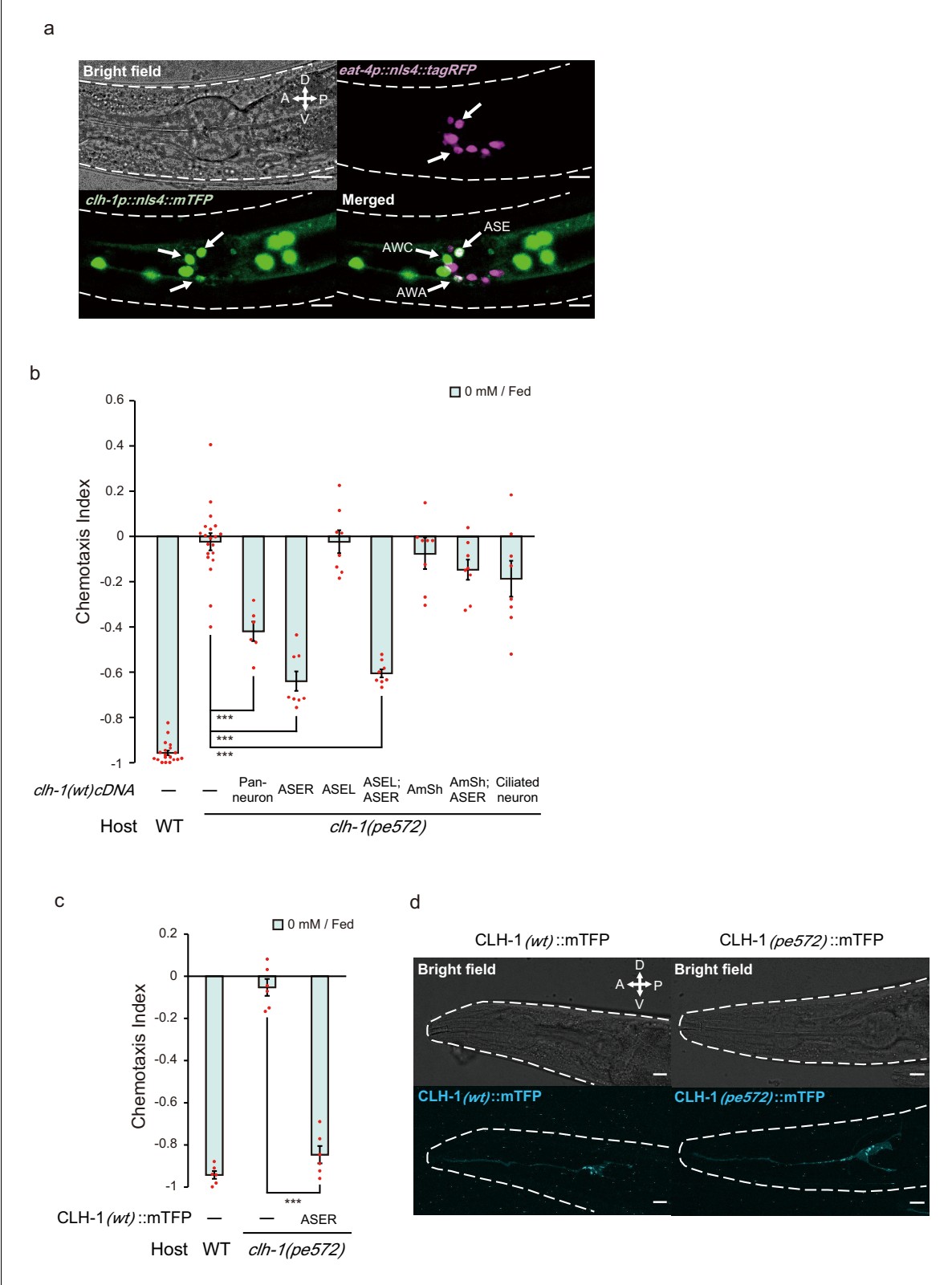

**Figure 3.** *clh-1* acts in the salt-sensing neuron ASER. (**a**) Expression pattern of *clh-1p::nls4::mTFP* (green, bottom left) in an adult animal. At least three pairs of sensory neurons, AWA, AWC and ASE expressed the marker. *eat-4p::nls4::tagRFP* and DiI, that marks glutamatergic neurons (magenta, top right) and six pairs of head sensory neurons (not shown), respectively, were used as position markers for cell identification. Scale bar = 10 μm. (**b**) Rescue of *clh-1(pe572)* mutants by cell-specific expression of *clh-1(wt)* cDNA. Promoters used in this experiment are as follows; *rimb-1p* for all neurons, *Figure 3 continued on next page*

*Figure 3 continued*

*gcy-5p* for ASER, *gcy-7p* for ASEL, *vap-1p* for amphid sheath cells, *dyf-11p* for ciliated neurons. Dots represent individual trials. Bars and the error bars represent mean +/- s.e.m., $n \geqq 6$ assays, Tukey's test. ***p<0.001. (c) Chemotaxis of *clh-1(pe572)* mutants that express *clh-1(wt)cDNA::mTFP* in ASER. mTFP-tagged CLH-1 is functional. Dots represent individual trials. Bars and the error bars represent mean +/- s.e.m., $n = 6$, Tukey's test. ***p<0.001. (d) Subcellular localization of CLH-1 in ASER. Panels show *gcy-5p::clh-1(wt)cDNA::mTFP* in wild type (left) and *gcy-5p::clh-1(pe572)cDNA::mTFP* in *clh-1 (pe572)* (right). Both CLH-1(*wt*)::mTFP and CLH-1(*pe572*)::mTFP localized to the membrane of dendrite, soma, axon, and cell organelles. Scale bar = 10 μm.

The online version of this article includes the following figure supplement(s) for figure 3:

**Figure supplement 1.** *clh-1* acts in ASER in salt chemotaxis, and mutations in *clh-1* do not affect morphology of ASER.

modulating efflux of Cl⁻ to the extracellular space of the receptive endings (*Singhvi et al., 2016*). In addition, it has been elucidated that CLH-1 transports anions to maintain intracellular pH of AmSh cells (*Grant et al., 2015*). These results implied a possibility that the receptive ending of ASER may be impaired in *clh-1(pe)* mutants. However, we did not find any notable change in the morphology of ASER including its sensory cilium (*Figure 3—figure supplement 1b–d*). Exposure to different salt and food conditions did not affect the length of ASER sensory cilium both in wild type and in *clh-1* mutants, suggesting that the morphology of ASER receptive ending remained largely unchanged (*Figure 3—figure supplement 1e*).

To examine subcellular localization of CLH-1 in ASER, we generated an mTFP-tagged CLH-1 (CLH-1::mTFP). The fusion protein was functional (*Figure 3c*). Fluorescent signals were detected in the plasma membrane and cell organelles (*Figure 3d*, left), as previously reported in mammalian ClC channels and ClC transporters, respectively (*Jentsch, 2008*; *Jentsch, 2007*). Localization patterns of CLH-1::mTFP with M293I mutation were comparable to those of wild type (*Figure 3d*, right). These results indicate that the mutation does not affect intracellular localization of CLH-1. It is unknown whether the CLH-1(*pe*) variants are expressed at higher or lower levels than CLH-1(*wt*), although expression level of a *clh-1* promoter-driven reporter slightly differed between genotypes (*Figure 5— figure supplement 1e*).

## Salt stimulus evokes flux of anions in ASER and this response is altered in *clh-1(pe)* mutants

Electrophysiological studies using *Xenopus* oocytes have shown that CLH-1 is an inwardly rectifying channel that conduct Cl⁻ and HCO3⁻, and it is activated by extracellular acidification (*Grant et al., 2015*; *Nehrke et al., 2000*). To examine whether the *clh-1(pe)* mutations affected the property of CLH-1 channel activity, we expressed mutant CLH-1 in *Xenopus laevis* oocytes and measured whole-cell membrane currents via two-electrode voltage clamping. Wild-type CLH-1 showed inwardly rectifying currents from −160 to −80 mV under the condition with pH 7.2 and 100 mM NaCl as previously reported (*Figure 4b and e*, referred to as pH 7 Cl⁻ condition, Materials and methods for details, *Grant et al., 2015*; *Nehrke et al., 2000*). Small, but similar trend of currents was observed in both CLH-1(M293I) and CLH-1(I146T) (*Figure 4c–e*). To further characterize the CLH-1 mutants, we measured currents under the conditions with extracellular pH reduced to 5.5 (pH 5 Cl⁻) or in the buffer in which 85 mM of chloride was replaced with an equimolar amount of bicarbonate (pH 7 HCO3⁻). The currents were increased by extracellular acidification in CLH-1 mutants as well as wild type (*Figure 4—figure supplement 1* and *Figure 4—figure supplement 2a–c*), indicating that sensitivity to pH is retained in the mutants. Also, regardless of expressed CLH-1 genotypes, the whole cell currents were comparable between pH 7 Cl⁻ and pH7 HCO₃⁻ conditions as previously reported (*Grant et al., 2015*, *Figure 4—figure supplement 1*). These results indicate that under our experimental conditions basic properties of CLH-1 as a voltage-dependent, extracellular pH-sensitive anion channel are retained in CLH-1(M293I) and CLH-1(I146T) mutants.

Next, we asked how *clh-1* mutations disturb salt chemotaxis of *C. elegans*. We focused on the salt-sensing neuron ASER, one of the site-of-action of *clh-1*, and which is essential for food-associated salt concentration chemotaxis (*Kunitomo et al., 2013*). ASER is activated by salt concentration decreases and deactivated by salt concentration increases (*Suzuki et al., 2008*, see below). Given that CLH-1 localized to the membranous compartments of ASER and acted as an anion channel, we hypothesized that CLH-1 would be involved in chloride dynamics during salt response of ASER. To

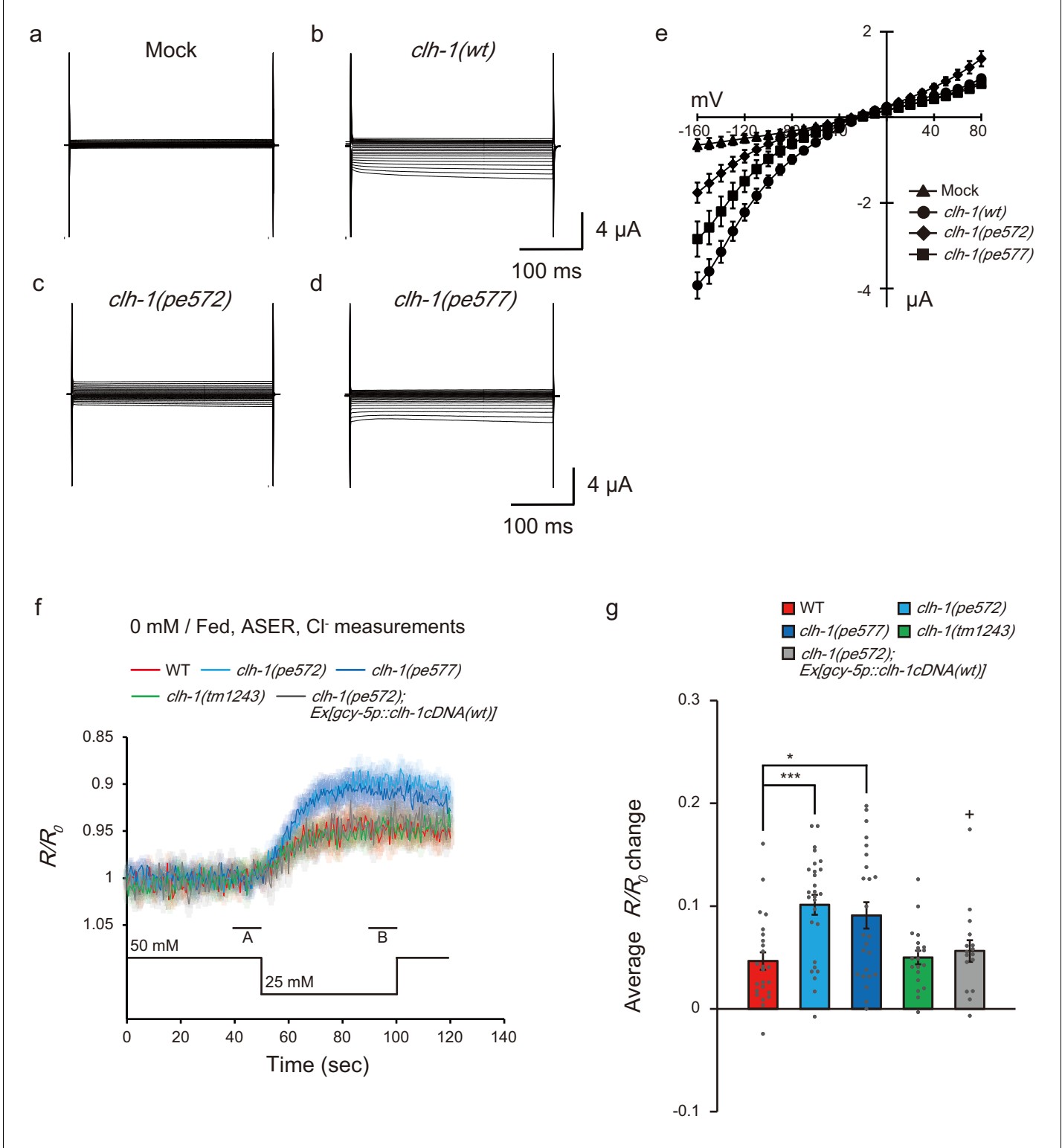

**Figure 4.** Mutations in *clh-1* affect chloride dynamics of ASER in response to salt down-step stimulus. (a–e) Representative current traces from *Xenopus* oocytes that expressed cRNA for mock (a), *clh-1(wt)* (b), *clh-1(pe572)* (c), and *clh-1(pe577)* (d) clamped at voltages ranging from −160 mV to 80 mV and perfused at pH 7 with Cl⁻ as the dominant extracellular anion. (e) The current-voltage relationships of mock (triangle, *n* = 7), *clh-1(wt)* (circle, *n* = 20), *clh-1(pe572)* (diamond, *n* = 13) and *clh-1(pe577)* (square, *n* = 16). The error bars represent s.e.m. (f) Responses of SuperClomeleon expressed in ASER after cultivation at 0 mM NaCl in the presence of food. External NaCl concentration was shifted from 50 mM to 25 mM at time 50 s. Note that the scale of the vertical axis is inverted so that increase in chloride concentration is displayed as up-shift of traces. A and B indicate the time points for calculating

*Figure 4 continued on next page*

*Figure 4 continued*

$R/R_0$ changes. The shaded region represents s.e.m., $n \geqq 17$ animals. (g) $R/R_0$ changes upon salt decrease. Mean +/- s.e.m., dots represent individual trials. $n \geqq 17$ animals, Tukey's test, ***p<0.001, *p<0.05, compared with wild type. +p<0.05, compared with *clh-1(pe572)* mutant.

The online version of this article includes the following figure supplement(s) for figure 4:

**Figure supplement 1.** Extracellular pH-sensitivity and bicarbonate permeability of CLH-1.

**Figure supplement 2.** Mutant CLH-1 channels retained pH sensitivity and bicarbonate permeability.

examine this possibility, we monitored intracellular chloride ($[Cl^-]_i$) dynamics of ASER by utilizing the genetically encoded chloride indicator, SuperClomeleon. This probe is a pH-sensitive FRET-type indicator in which YFP/CFP fluorescence ratio decreases upon binding to chloride ions (*Arosio and Ratto, 2014*; *Grimley et al., 2013*, see Discussion). Animals were cultivated at 0 mM NaCl with food and immobilized in a microfluidics device (*Chronis et al., 2007*), and a NaCl down-step from 50 mM to 25 mM was applied as salt stimulus. In wild type, the YFP/CFP ratio decreased upon salt down-step, indicating that $[Cl^-]_i$ of ASER was increased when the neuron was activated (*Figure 4f*). This response is not, at least solely, mediated by CLH-1 because it was also observed in *clh-1(tm1243)* mutants. Notably, the magnitude of YFP/CFP ratio change was significantly larger in *clh-1(pe)* mutants, suggesting that ASER $[Cl^-]_i$ greatly increased in the mutants (*Figure 4f,g*). The enhanced $[Cl^-]_i$ of *clh-1(pe572)* was restored by expression of *clh-1(wt)* cDNA in ASER, indicating that CLH-1 acts cell-autonomously in regulation of $[Cl^-]_i$ during depolarization.

## ASER salt response is altered in *clh-1* mutants

Intracellular chloride concentration increase can antagonize depolarization of neurons. We then wanted to look into whether *clh-1* mutations affect the activity of ASER in response to salt stimulus. As aforementioned, ASER is activated by salt concentration decreases, which is indicated by an increase in intracellular calcium levels, whereas it is deactivated by salt concentration increases (*Suzuki et al., 2008*). Such ASER responsivity is basically retained regardless of cultivation salt concentrations or food availability (*Kunitomo et al., 2013*; *Oda et al., 2011*).

We performed in vivo calcium imaging in wild type and *clh-1* mutants using a genetically encoded calcium indicator YC2.6 (*Chronis et al., 2007*; *Horikawa et al., 2010*). Animals were cultivated at either 0 mM NaCl or 100 mM NaCl with or without food, and ASER was stimulated by repeated salt concentration changes from 50 mM NaCl to 25 mM NaCl, to observe responses to both down-step and up-step stimuli. After cultivation at 0 mM NaCl, the amplitude of calcium response to the first down-step stimulus was comparable between wild type and *clh-1* mutants. However, the response to the second down-step was diminished in *clh-1(pe)* mutants compared to wild type (*Figure 5a,b*). A similar trend was observed in the third down-step response. The decay of intracellular calcium level ($[Ca^{2+}]_i$) was small in the *clh-1(pe)* mutants, which was more evident after salt up-step (*Figure 5—figure supplement 1a,b*). This diminished decay was likely responsible for the decreased calcium response to the repeated stimuli. The delayed response of *clh-1(pe572)* mutant was rescued by expression of *clh-1(wt)* cDNA in ASER (*Figure 5a,b* and *Figure 5—figure supplement 1a,b*). On the other hand, ASER salt response of *clh-1(tm1243)* was similar to that of wild type, except that the decay was significantly large during the first down-step stimulus. Interestingly, reduction of ASER response amplitude upon repeated salt down-step stimuli was not obvious after cultivation at 100 mM NaCl with food, although *clh-1(pe572)* constantly showed small ASER responses (*Figure 5c,d* and *Figure 5—figure supplement 1c,d*). Considering the essential role of ASER in salt concentration chemotaxis, these results imply that hampered chemotaxis of *clh-1(pe)* mutants toward low salt is probably due to the abnormal ASER responsivity to salt concentration changes. We observed expression of a *clh-1* promoter-driven transcriptional reporter in ASER and found a reduction of its expression in wild type, but not in the *clh-1(pe)* mutants, after cultivation at 0 mM NaCl (*Figure 5—figure supplement 1e*). This result raises a possibility that the expression change of CLH-1 in ASER might contribute to the difference of chloride and calcium responses of the cell between wild type and the *clh-1(pe)* mutants.

Consistent with the previous reports, salt responses of ASER of starved wild-type animals were not largely different when compared to that of fed animals. However, the activity patterns of *clh-1* mutants starved at 0 mM were distinct from those of fed animals (please compare *Figure 5*,

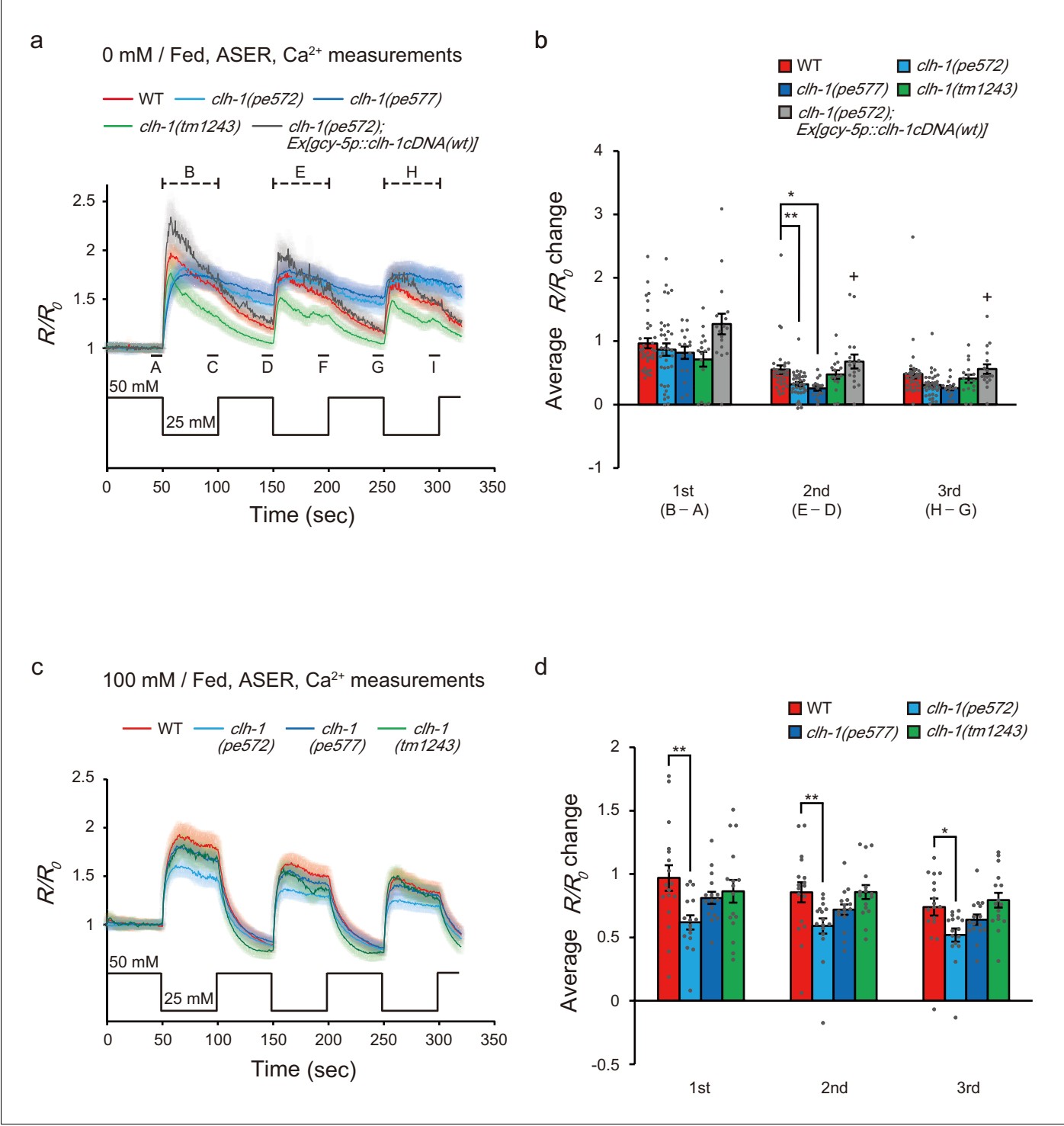

**Figure 5.** Calcium dynamics of ASER in response to repeated salt stimuli is altered in *clh-1* mutants. (**a and c**) Calcium responses of ASER stimulated by NaCl concentration changes between 50 mM and 25 mM after cultivation at 0 mM NaCl (**a**) or 100 mM NaCl (**c**) in the presence of food for 6 hr. A to I indicate the time points for calculation of $R/R_0$ changes. A, D, and G are the time points for pre-stimulus $R/R_0$, B, E, and H are the time points for peak $R/R_0$ during stimulation, C, F, and I are the time points for decayed $R/R_0$ during stimulation. The shaded region represents s.e.m., $n \geqq 16$ animals. (**b and d**) $R/R_0$ changes at each NaCl down-step stimulus (B - A, E - D, and H - G for the 1st, 2nd, and 3rd stimulus, respectively). 0 mM NaCl cultivated (**b**) or 100 mM cultivated (**d**). See Materials and methods for details. Bars and the error bars represent mean +/- s.e.m., dots represent individual trials. $n \geqq$ 16 animals, Tukey's test, **$p<0.01$, *$p<0.05$. +$p<0.05$, compared with *clh-1(pe572)* mutant.

The online version of this article includes the following figure supplement(s) for figure 5:

*Figure 5 continued on next page*

*Figure 5 continued*

**Figure supplement 1.** *clh-1* missense mutations affect calcium dynamics of ASER.
**Figure supplement 2.** Effect of *clh-1(pe)* mutations on calcium dynamics of ASER in starved animals.

*Figure 5—figure supplement 1* and *Figure 5—figure supplement 2*). After cultivation at 0 mM without food, for example, the amplitude of activation of *clh-1(pe577)* mutants were even larger than that of wild type (*Figure 5—figure supplement 2a–c*). Meanwhile, the difference of ASER salt responses between wild type and *clh-1(pe)* mutants were less obvious after starvation at 100 mM NaCl (*Figure 5—figure supplement 2e–h*). Taking all these into account, we concluded that mutations in *clh-1* affect responsivity of ASER, most notably in the reduced response to repeated salt down-step and up-step stimuli after cultivation at 0 mM NaCl with food.

## Behavioral strategies for chemotaxis are disrupted in *clh-1(pe)* mutants

Next, we quantitatively analyzed the navigation behavior of *clh-1* mutants to examine which behavioral components are affected. *C. elegans* utilize at least two behavioral strategies to achieve salt chemotaxis: klinokinesis and klinotaxis. In klinokinesis, migration bias is generated by controlling the frequency of steep turns called pirouettes, which are typically accompanied by reversals and omega turns. The bout of pirouette is triggered according to cumulative salt concentration change along an animal's progression (*Pierce-Shimomura et al., 2001*). In klinotaxis, animals gradually curve toward their preferred direction by sensing fluctuation of salt concentration accompanying head bending (*Iino and Yoshida, 2009*). Input to ASER is both required and sufficient for fed animals to generate the two behavioral strategies (*Kunitomo et al., 2013*; *Satoh et al., 2014*). We found that klinotaxis bias was severely impaired in *clh-1(pe)* mutants regardless of cultivation salt concentrations (*Figure 6a* and *Figure 6—figure supplement 1a,c*). Klinokinesis bias of the *clh-1(pe)* animals after cultivation at 100 mM NaCl was comparable to that of wild type (*Figure 6b* and *Figure 6—figure supplement 1d*). These results suggest that chemotaxis of *clh-1(pe)* animals to high salt after cultivation at 100 mM NaCl largely depends on klinokinesis. However, after cultivation at 0 mM NaCl, klinokinesis bias was lost in *clh-1(pe)* mutants. Up-regulation of pirouette frequency along with salt concentration increase ($dC/dt > 0$) was abolished in *clh-1(pe)* mutant animals (*Figure 6b* and *Figure 6—figure supplement 1b*). These results strongly indicate that the defective chemotaxis of *clh-1(pe)* mutants is due to loss of both klinotaxis and klinokinesis biases after cultivation at 0 mM NaCl. The klinokinesis and klinotaxis of *clh-1(tm1243)* mutants were comparable to those of wild type (*Figure 6a,b* and *Figure 6—figure supplement 1a,b*).

## Reduced salt response of AIB to salt increase in *clh-1(pe572)* mutants

To further gain insight into the neural mechanism of klinokinesis defects of the *clh-1(pe)* mutants after cultivation at 0 mM NaCl, we focused on AIB, a postsynaptic interneuron of ASER which promotes sensory stimulus-dependent reversals, the trigger of pirouettes (*Piggott et al., 2011*; *Zou et al., 2018*). Because the synapse between ASE and AIB is proposed as the site of plasticity that regulate klinokinesis bias in salt chemotaxis (*Kunitomo et al., 2013*; *Luo et al., 2014*; *Wang et al., 2017*), we hypothesized that the responsivity of AIB may be altered in these mutants. To examine this possibility, we observed salt responses of AIB in freely behaving animals using a microfluidics arena (*Albrecht and Bargmann, 2011*). Animals were cultivated at 0 mM NaCl with food and stimulated by an up-step NaCl stimulus from 0 mM to 25 mM. Wild-type animals responded to the salt stimulus by slowing down or reversal (the moving velocity becomes less than zero, *Figure 6c* and *Figure 6—figure supplement 2*). *clh-1(pe572)* mutants also showed reduction in speed, but the proportion of animals that exhibited reversal was smaller than wild type (*Figure 6d*). These results agreed with the klinokinesis defect observed on chemotaxis plates (*Figure 6b* and *Figure 6—figure supplement 1b*). Importantly, responses of AIB to salt up-step correlated well with behaviors. AIB was largely activated upon salt stimulus in wild type, which was absent in *clh-1(pe572)* mutants (*Figure 6e,f*). Expression of *clh-1(wt)* cDNA in ASER restored both reversal behavior and AIB response, indicating that malfunction of *clh-1* in ASER caused the defects. Collectively, our results indicate that impaired salt responses of ASER-AIB salt circuit resulted in

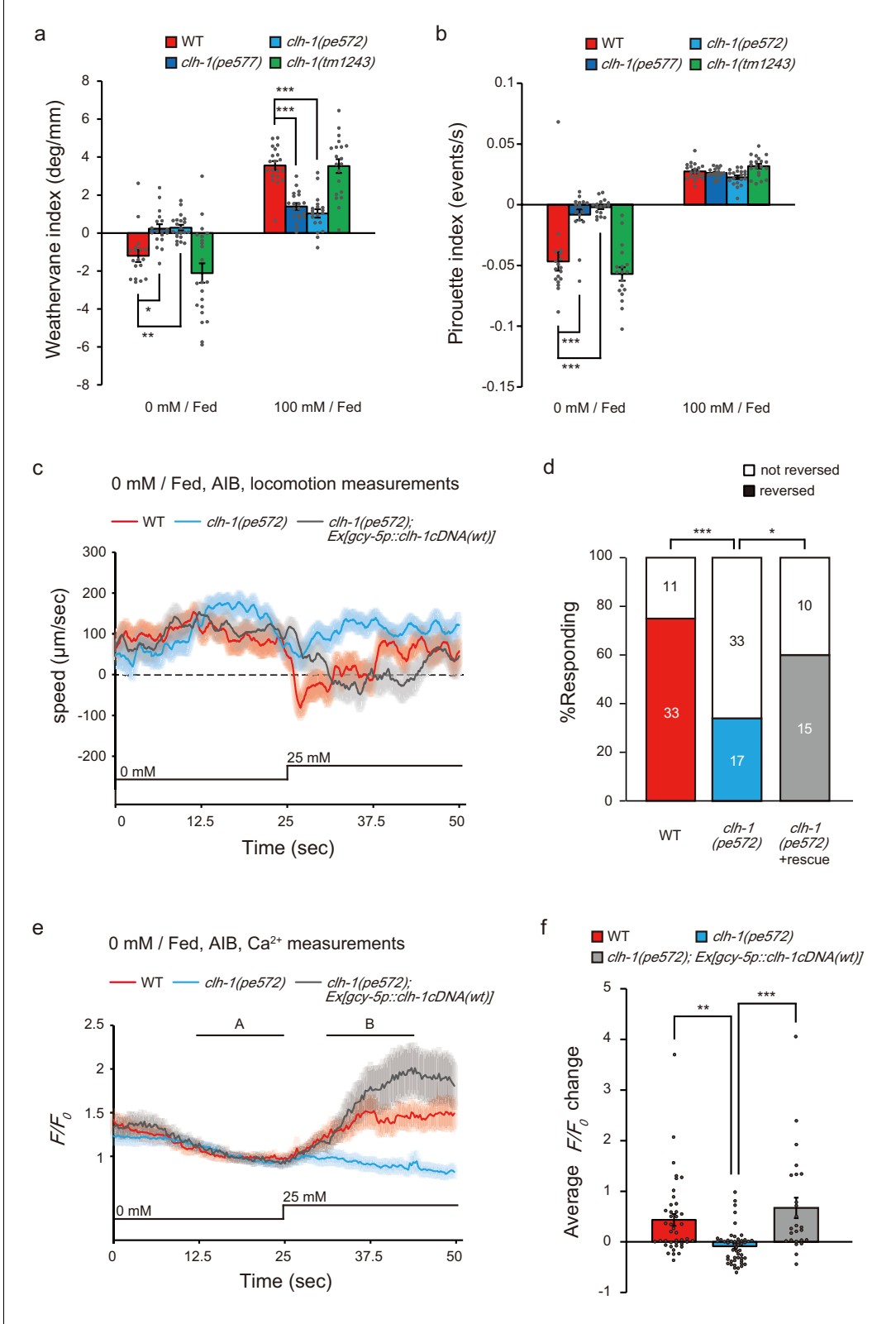

**Figure 6.** Missense mutations in *clh-1* attenuate both klinotaxis and klinokinesis, as well as AIB response and reversal in response to salt increase. (a and b) *clh-1(pe)* mutants show defects in migration bias in salt chemotaxis. Bias of klinotaxis (a) and klinokinesis (b), represented by weathervane index and pirouette index, respectively. In both mechanisms, positive and negative values indicate migration bias toward higher and lower salt concentrations, respectively. Bars and the error bars represent mean +/- s.e.m., dots represent individual trials. *n* ≧ 18 assays, Dunnett's test,

*Figure 6 continued on next page*

*Figure 6 continued*

***p<0.001, **p<0.01, *p<0.05. (**c and e**) Locomotion speed of animals (**c**) and calcium responses of AIB (**e**) after cultivation at 0 mM NaCl in the presence of food. In panel (**e**), A and B indicate the time points for calculation of $F/F_0$ changes. NaCl concentration change from 0 mM to 25 mM at 25 s. The shaded region represents s.e.m., $n \geqq 25$ animals. (**d**) Proportion of animals that showed reversal after salt stimulus. Reversal was defined as follows; backward locomotion, whose velocity less than $-100$ µm/sec was continued for more than 1 s (35 frames). The error bars represent s.e.m., $n \geqq 25$ animals, Fisher's exact test. ***p<0.001, *p<0.05. (**f**) $F/F_0$ change upon salt stimulus (B - A, see Materials and methods for details). Bars and the error bars represent mean +/- s.e.m., dots represent individual trials. $n \geqq 25$ animals, Tukey's test, ***p<0.001, **p<0.01.

The online version of this article includes the following figure supplement(s) for figure 6:

**Figure supplement 1.** Quantification of the navigation behaviors.

**Figure supplement 2.** Quantification of reversal behaviors in microfluidics arena.

reduction of turning frequency in *clh-1(pe572)* mutants upon salt up-step after cultivation at 0 mM NaCl.

## Discussion

Here, using genetic, neurophysiological, and behavioral analyses, we described a neural mechanism in which novel missense mutations of the ClC chloride channel *clh-1* disturb responses of a sensory neuron and eventually alter animal's behavior in *C. elegans*. This is probably carried out by impairment of, (i) ASER responsivity to repeated salt stimuli that affect klinotaxis and (ii) salt up-step response of ASER and thereby of AIB after cultivation at low salt that contribute to klinokinesis (*Figure 7*). Repeated activation of ASER synchronized with head swing generates biased klinotaxis (*Satoh et al., 2014*). Reduced responsivity to repeated salt stimuli in *clh-1(pe)* mutants implied that temporal resolution of ASER is impaired in the mutants (*Figure 5a–b* and *Figure 5—figure supplement 1a*). Klinotaxis was actually disrupted in the mutants regardless of previous cultivation conditions (*Figure 6a* and *Figure 6—figure supplement 1a,c*). Our data is consistent with the idea that dynamic $[Ca^{2+}]_i$ fluctuation in ASER reflecting environmental NaCl concentration change is required for generation of klinotaxis. In addition, *clh-1(pe)* mutants showed klinokinesis defect upon salt increase after cultivation at 0 mM NaCl (*Figure 6b* and *Figure 6—figure supplement 1b*). In agreement with this, responses of ASER and AIB, and reversal behavior upon salt up-step were reduced in *clh-1(pe572)* animals (*Figure 5a* and *Figure 6d,f*). Suppression of AIB activity results in reduction of turning frequency (*Gordus et al., 2015*; *Piggott et al., 2011*). Unresponsiveness of AIB to salt increase will therefore result in reduced migration toward low salt regions. Defects in migration to low salt can explain the increased salt preference of the *clh-1(pe)* mutants after cultivation at 50 mM NaCl (*Figure 1b*). Wild-type animals wander around the central area on chemotaxis assay plate by suppressing migration toward both lower and higher salt concentrations after cultivation at 50 mM NaCl (*Kunitomo et al., 2013*; *Ohno et al., 2014*). Therefore, the loss of migration bias toward low salt under such condition would drive animals toward high salt.

Monitoring of $[Cl^-]_i$ using SuperClomeleon indicated an influx of chloride into ASER upon activation of the cell. In general, influx of anion prevents depolarization in neurons (*Staley et al., 1995*). There are several possible mechanisms that could explain increased chloride influx by *clh-1(pe)* mutations (*Figure 7*, middle). One is an elevated anion intrusion via CLH-1(pe) channels. ClC-2, the closest mammalian homolog of CLH-1, was shown to be involved in chloride influx into neuronal cells, probably due to incomplete rectification (*Ratté and Prescott, 2011*; *Rinke et al., 2010*). Another possible mechanism is a difference in ASER's transmembrane chloride potential between genotypes. ClC proteins, as well as other chloride transporters such as $K^+/Cl^-$ cotransporters (KCCs) and $Na^+/K^+/Cl^-$ cotransporters (NKCCs), are involved in the excitability of neurons through their homeostatic roles in regulating cellular ionic milieu (*Jentsch, 2008*; *Stauber et al., 2012*). It remains unclear how extracellular chloride of ASER is regulated. In this study, we showed that *clh-1(wt)* cDNA failed to rescue *clh-1(pe572)* phenotype if expressed in AmSh or ciliated neurons together with ASER (*Figure 3b*). This result suggests that CLH-1 may differently act depending on cell type. Indeed, expression of a $K^+/Cl^-$ cotransporter KCC-3 is restricted in glial cells including AmSh (*Katz et al., 2019*; *Spencer et al., 2011*; *Wallace et al., 2016*). KCC-3 is involved in responsivity of the thermosensory neuron AFD through regulating extracellular chloride level of the sensory endings (*Singhvi et al., 2016*; *Yoshida et al., 2016*). Since chloride milieu of a cell is maintained by a balance

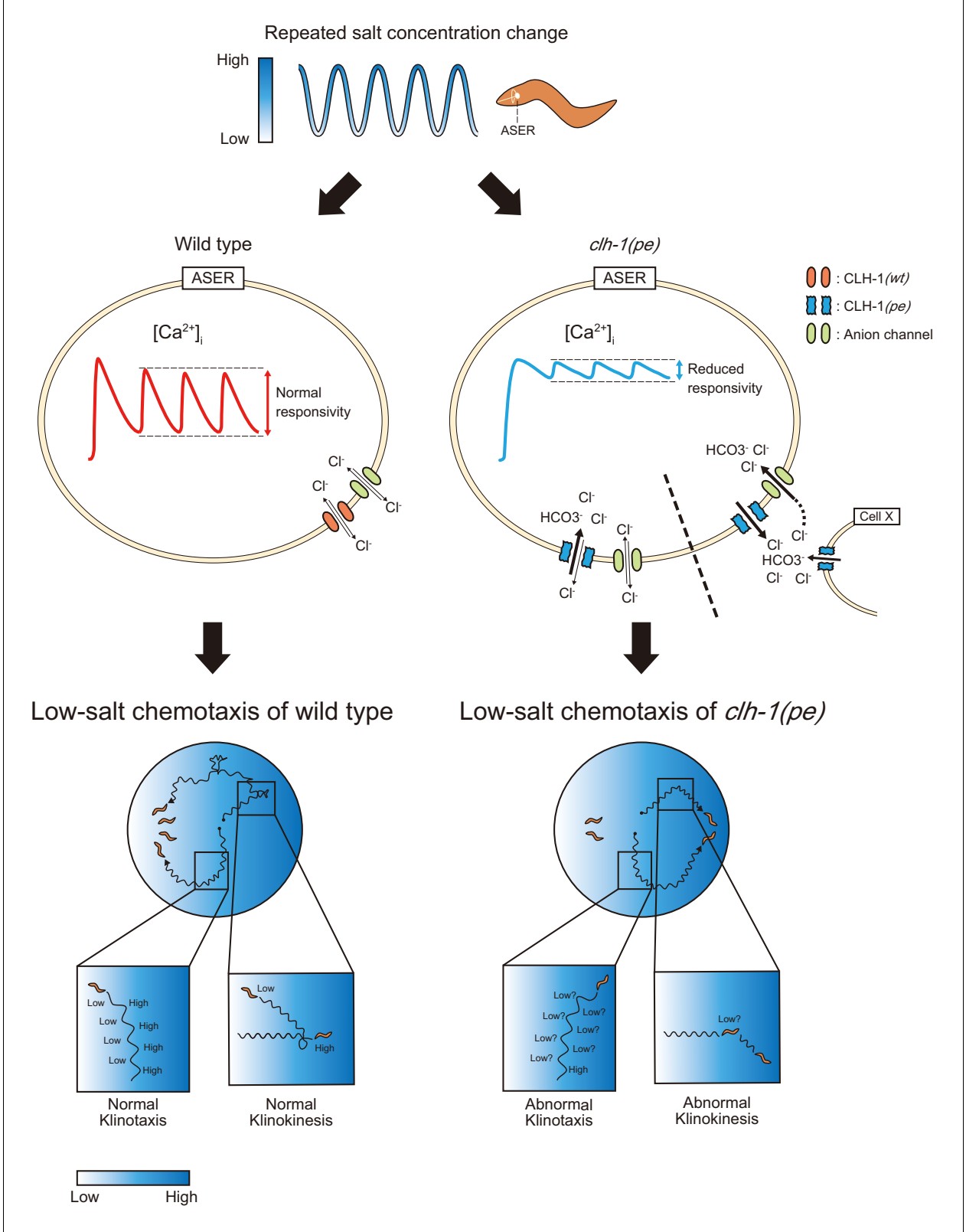

**Figure 7.** A model for the mechanism of impaired low salt chemotaxis of *clh-1(pe)* mutants. Animals experience salt concentration fluctuation along with migration on salt gradient, which is depicted by blue waves (top). Salt information is sensed by the ASER neuron and translated into intracellular calcium transients (middle). In the *clh-1(pe)* mutants, calcium responses of ASER to repeated salt stimuli is diminished after cultivation at low-salt concentration. This is probably due to increased influx of chloride ions into the cell upon depolarization. Permeation of bicarbonate ions through CLH-1

*Figure 7 continued on next page*

Figure 7 continued

may also be involved. Although the mechanism for increased chloride influx is currently unknown, we suggest two possible scenarios; increased anion influx through CLH-1(*pe*) channels (left of the dashed line) and elevated extracellular chloride level (right of the dashed line). Reduced responsivity of ASER then results in hampered migration bias toward low salt both in klinokinesis and klinotaxis (bottom).

of these channels and transporters activity, the aforementioned chloride regulation could be disturbed when CLH-1 was chronically overexpressed in AmSh.

SuperClomeleon is an FRET-based sensor sensitive to pH change (*Arosio and Ratto, 2014*; *Grimley et al., 2013*). In addition, activity-dependent cytoplasmic acidification is reported in *C. elegans* sensory neurons (*Ventimiglia and Bargmann, 2017*). As we observed the SuperClomeleon responses upon activation of ASER, there is a possibility that cytoplasmic pH change confounded the measurement of chloride concentration. Furthermore, CLH-1 is $HCO_3$-permeable and contributes to cellular pH regulation (*Grant et al., 2015*). Hence, pH regulation might be altered in *clh-1 (pe)* mutants in vivo (*Figure 7*, middle right). These scenarios, together with further characterization of channel properties of CLH-1, needs to be demonstrated in the future study.

In the levamisole resistance test, missense (*pe572* and *pe577*) and the deletion (*tm1243*) allele of *clh-1* showed an opposite phenotype (*Figure 1—figure supplement 3c*). In addition, overexpression of *clh-1*(*wt*) impaired low-salt chemotaxis, which partially resembled the phenotype of *clh-1*(*pe*) mutants (*Figure 1c*). These results suggest that the *clh-1*(*pe*) mutations are neomorphic or perhaps hypermorphic alleles. ClC proteins act in homo- or heterodimers (*Stölting et al., 2014*). It is not predictable whether CLH-1(*pe*)/CLH-1(*wt*) heterodimer shows mutant-type or wild-type activity. The recessiveness of *clh-1*(*pe*) mutations (*Figure 1d*) can be explained by assuming that the heterodimer shows wild-type activity. Although genetic analyses indicated that the mutations of *clh-1* affected the property of CLH-1 channels, we did not find a remarkable change in the electrophysiological analyses except that currents were smaller in the mutants. Limitations of using heterologous system may explain such disagreement (reviewed in *Bianchi, 2006*; *Dascal, 2000*). First, cellular environment of *Xenopus* oocyte differs from that of *C. elegans* neurons. Inward currents were observed in membrane potential range as low as −80 mV or lower in electrophysiology, whereas responses of ASER was observed under depolarized conditions (*Shindou et al., 2019*). In addition, modification of CLH-1 proteins or interaction of CLH-1 with other proteins may be absent in *Xenopus* oocyte. The CBS domain is known to regulate channel activity depending on phosphorylation or binding to auxiliary subunits (*Denton et al., 2005*; *Estévez et al., 2004*; *Stölting et al., 2013*). Such modifications may also occur in CLH-1. Regarding the amplitude of whole-cell membrane current, the expression level of channel proteins might have been different between wild type and mutants. Stability of channel proteins also could be affected by the mutations. To date, substantial number of mutations have been characterized in ClC genes, most of which are mammalian ClCs. To our knowledge, however, none of these mutations were similar to *clh-1*(*pe*) mutations. Future studies such as single-channel analyses are required to fully characterize the property of mutant CLH-1 channels.

ASER makes synapses to three first layer interneurons, AIA, AIB and AIY. Activation of AIA and AIY are known to promote forward locomotion, while activation of AIB promotes reversals (*Gray et al., 2005*; *Piggott et al., 2011*; *Zou et al., 2018*). The salt response of AIB is markedly changed by salt experience. AIB is activated by salt concentration decrease if previously-experienced salt concentrations were high, whereas it is rather deactivated after cultivation at low salt (*Kunitomo et al., 2013*; *Luo et al., 2014*). Here, we showed that AIB is activated by salt increase after cultivation at low salt, which was reduced in *clh-1*(*pe572*) mutant (*Figure 6e,f*). This response is at least partially dependent on ASER because AIB response was restored by expression of *clh-1*(*wt*) in ASER. Considering that ASER is glutamatergic (*Serrano-Saiz et al., 2013*), there may be both excitatory and inhibitory transmission between ASER and AIB. Indeed, AIB expresses the AMPA-type glutamate receptor GLR-1, which mediates excitatory glutamatergic inputs from sensory neurons (*Chalasani et al., 2007*; *Zou et al., 2018*). Besides, it has recently been suggested that the glutamate-gated chloride channel GLC-3 and AVR-14 would mediate inhibitory inputs to AIB (*Kuramochi and Doi, 2018*; *Summers et al., 2015*). These observations raise the possibility that glutamate response of AIB depends on the electrochemical gradient of Cl$^-$ across the membrane as well as the balance of excitatory and inhibitory receptors. In other words, under the condition in which inhibitory glutamate receptors were dominant, AIB could be disinhibited by reduction of

presynaptic glutamate release. Growing evidence highlights the importance of chloride homeostasis in the function of the nervous system. In mammalian hippocampal neurons, intracellular chloride is elevated during embryonic development and thereby renders GABAergic transmission excitatory, which is necessary for maturation of synaptic network (*Pfeffer et al., 2009*). Furthermore, chloride transportation through NKCC1 regulates synaptic plasticity and memory formation in adult hippocampal neurons (*Deidda et al., 2015*). It will be of interest in the future to determine how the extracellular ionic milieu and glutamate receptors orchestrate the responsiveness of AIB.

In this study, we showed that ClC channel/transporter genes function redundantly in salt chemotaxis of *C. elegans*. Food-associated salt chemotaxis was normal in each single (*clh-1(tm1243)*, *clh-2 (ok636)*, *clh-3(ok763)*, or *clh-4(ok1162)*) or *clh-2(ok636) clh-1(tm1243)* double mutants. However, lack of three ClC channels, *clh-3(ok763) clh-2(ok636) clh-1(tm1243)* triple mutation, gave a marked effect on behavior (*Figure 2—figure supplement 1b* and *Figure 2b*). Interestingly, these three genes are located very closely on chromosome II (4.08 +/- 0.003 cM for *clh-1*, 3.46 +/- 0.003 cM for *clh-2*, 0.50 +/- 0.000 cM for *clh-3*), suggesting that they are derived by duplication and they might share some evolutionarily conserved functions. The *clh-5* putative anion transporter gene is also located on chromosome II: 1.01 +/- 0.007 cM. *clh-5(tm6008)* and many of the *clh* multiple mutants that carry *clh-5 (tm6008)* showed high immobility indices (*Figure 2—figure supplement 1b,c* and *Figure 2—figure supplement 1a*), suggesting that the mutants displayed a locomotion defect on chemotaxis assay plate. Locomotion defects can interfere with chemotaxis of the animals since it prevents prompt migration toward preferred conditions. Thus, improvement of locomotion might ameliorate the chemotaxis defects in these mutants. Actually, mutation of *clh-3* partially suppressed chemotaxis defect of the *clh-5 clh-2 clh-1* triple mutation (that is, in the *clh-5 clh-3 clh-2 clh-1* quadruple mutants) as well as locomotion defects. These results imply that *clh-5* and *clh-3* are involved in locomotion upon chemotaxis. Mutation of *clh-6(tm617) V*, which encodes a putative anion transporter and whose single mutation had no effect on salt chemotaxis, also gave rise to salt chemotaxis defect in combination with *clh-2(ok636) clh-1(tm1243) II* mutations.

As we did not examine all combinations of multiple loss-of-function mutation but started with the *clh-2(ok636) clh-1(tm1243)* double mutant, it is unclear whether *clh-1* is naturally involved in salt chemotaxis. Although loss of *clh-1* showed no remarkable salt chemotaxis defects, it affected salt responses of ASER under starvation conditions (*Figure 5—figure supplement 2*). Function of the sensory neurons other than ASER are required to properly avoid the salt concentration of starved conditions (*Jang et al., 2019*; *Watteyne et al., 2020*). Cooperation of multiple sensory neurons might have compensated for the defective salt response of ASER. Intriguingly, animals that lacked all ClC genes were viable, but showed severe salt chemotaxis defect. Only a few studies so far have addressed functional redundancy of ClC family proteins in an organism (*Jeworutzki et al., 2014*; *Stölting et al., 2014*), and our study provides an insight into functional differences and redundancies of ClC family proteins.

## Materials and methods

### *C. elegans* strains and culture

Bristol N2 was used as wild-type *C. elegans*. All mutant strains were outcrossed multiple times with N2. *E. coli* NA22 was used as a food source for behavioral analyses including salt chemotaxis assay. For imaging experiments, OP50 was used as a food source. Strains used in this study are listed in *Supplementary file 1*.

### Behavioral tests

Salt chemotaxis assays were performed as previously reported with minor modifications (*Kunitomo et al., 2013*). Chemotaxis assay plate was prepared as follows. On top of an agar plate (2% agar, 25 mM potassium phosphate (pH 6.0), 1 mM $CaCl_2$, 1 mM $MgSO_4$, 8.5 cm in diameter and 1.76 mm in thickness with 10 mL agar), NaCl gradient was created by placing two cylindrical agar blocks (14.5 mm in diameter and 5.3 mm in thickness) that contained 0 mM (position A) or 150 mM (position B) of NaCl in the composition of background plate. The agar blocks were removed after 18 to 20 hr, just before assay. Animals were cultivated on regular nematode growth medium (*Brenner, 1974*) to young adults, and further cultivated on NGM plates that contained either 0 mM or

100 mM NaCl in the presence or absence of food for pre-assay cultivation for 6 hr. Fifty to 200 worms were collected, washed twice with wash buffer (50 mM NaCl, 25 mM potassium phosphate (pH 6.0), 1 mM CaCl$_2$, 1 mM MgSO$_4$, 0.02% gelatin), then placed at the center of the assay plate. Animals were allowed to run for 45 min. One microliter each of 0.5 M NaN$_3$ was spotted to the position A and B so that worms that had reached to these positions were immobilized. The number of worms ($N$) within area A and B (a 2 cm radius from the center of each agar cylinder's position A and B) and area O (ellipse with radii 20 mm and 10 mm around the start point) as well as the total number of worms on assay plate were counted to calculate the chemotaxis index and immobility index as follows (*Figure 1—Figure 1—figure supplement 1a*).

$$Chemotaxis\ index = (N_B - N_A)/(N_{Total} - N_O)$$
$$Immobility\ index = N_O - N_{Total}$$

The values of chemotaxis index 1.0 and −1.0 represent complete migration toward high and low salt regions, respectively, whereas zero value represents no preference for salt concentration (unbiased migration) or a preference for concentration near the central region. Levamisole resistance was determined on agar plates that contained 0.5 mM drug at room temperature. Body paralysis was defined as the lack of body movement in response to prodding by toothpick by visual inspection every 15 min (*Gottschalk et al., 2005*; *Lewis et al., 1980*).

## Forward genetic screening and identification of the responsible gene

Wild-type animals were mutagenized with ethyl methanesulfonate (EMS) as described (*Brenner, 1974*). Progenies were tested for food-associated salt concentration chemotaxis and mutants were selected; animals that showed defective chemotaxis were collected from the assay plate and propagated for testing at the next generation. For detail, we collected animals that approached high-salt region (the region B in *Figure 1—figure supplement 1a*) after cultivation at 0 mM NaCl (*Figure 1—figure supplement 1b*). This strategy could yield two types of mutant candidates; ones that preferred high-salt concentration and ones that show unbiased preference to salt concentration. The latter tend to randomly distribute on salt gradient and therefore could come to high-salt region by chance. JN572 and JN577 showed an unbiased salt chemotaxis after cultivation at 0 mM with food, but they showed an obvious preference to high salt after cultivation at 50 mM or higher, demonstrating that the mutants are intrinsically capable of chemotaxis. This process was repeated until F6. We screened approximately 24,000 F2 animals and obtained seven independent mutants. After being outcrossed with N2, two mutant strains JN572 and JN577 were further analyzed. Other mutants will be described elsewhere.

We mapped the responsible gene for JN572 and JN577, each respectively appeared to be *clh-1 (pe572)* and *clh-1(pe577)* (see text), by using single nucleotide polymorphisms (SNPs) between N2 and CB4856 (*Fay and Bender, 2006*; *Wicks et al., 2001*) Both of these mutations were mapped between 2.82 (SNP: *WBVar00175127*) and 6.12 (SNP: *WBVar00176673*) on chromosome II. Genome sequences identified a missense mutation in *clh-1* in each strain. We performed rescue experiments using fosmids and genomic PCR fragment of candidate genes. Both mutants were successfully rescued by WRM0612bF09 (fosmid) or genomic PCR fragment of *clh-1*/T27D12.2. Thus we concluded *clh-1* was the causative gene. *pe572* was a G to A transition, whose 5′ and 3′ flanking sequences are TGCACATTCTCGGCGCCTAT and GGAGGTAGGGCTTAACCCTT, respectively. *pe577* was a T to C transition, whose 5′ and 3′ flanking sequences are GATTTTCATCGATATGGGAA and TGAGTATCTGATTCATTGTG, respectively.

## Genotyping

Alleles of each gene locus were verified by PCR using sequence-specific primers for the target sequences. Genotyping primers used in this study are listed in *Supplementary file 2*.

## *clh-1* expression constructs

Full length *clh-1a*/T27D12.2a cDNA clones for wild type, *clh-1(pe572)* and *clh-1(pe577)* mutants were obtained by RT-PCR using sequence-specific primers (5′- GCTAGCCAGGATGGAAGACGCCG TCGTCGT −3′ and 5′- GGTACCTTAGCGGGTTTCGTCATCCG −3′). PCR products were cloned to the pDEST vector (Invitrogen) as a *Nhe*I-*Kpn*I fragment, and whose sequence was confirmed.

*clh-1* genomic DNA fragments (*clh-1gDNAs*) were prepared by PCR using sequence-specific primers (5′- ATTGCACACATAATTGCGGTAGAC −3′ and 5′- TTGACCCATAAGGTGTAGGCTGC −3′). The nucleotide sequence of the open-reading frame was confirmed after cloning.

## Plasmid construction

Vectors for cell-specific expression in *C. elegans* were generated using the Gateway cloning system (Invitrogen). For *clh-1* promoter, a 7.5 kb DNA fragment that contained 4.0 kb upstream of transcription initiation site and 3.5 kb downstream of the first exon of *clh-1a* were amplified from *C. elegans* genome using primers: 5′- GCACACATAATTGCGGTAGAC −3′ and 5′- CGCATTTTCTTGAACCC TGG −3′ and cloned into an entry vector, pENTR-1A. For *vap-1* promoter, which specifically expresses in amphid sheath glial cells (*Bacaj et al., 2008*), we amplified 2.5 kb upstream of the first exon of *vap-1* using primers: 5′- ATTTATAGAAAGTTTCCAAA −3′ and 5′- CTGTGAAAATGAACG-CACGC −3′. pDEST-*SL2::nls4::mTFP* was generated by ligating the trans-spliced leader sequence SL2, four repeats of the nuclear localization signal, and the fluorescence protein mTFP on a pDEST vector. For pDEST-*clh-1cDNA::mTFP*s, *KpnI-EcoRV* fragment from pDEST-*mTFP* was cloned into the pDEST-*clh-1cDNA* vectors. SuperClomeleon (*Grimley et al., 2013*) was codon-optimized for expression in *C. elegans* using Codon adapter (*Redemann et al., 2011*), synthesized, and cloned into pDEST vector. The expression constructs pG-*clh-1p::nls4::mTFP*, pG-*rimb-1p::clh-1(wt)cDNA*, pG-*gcy-5p::clh-1(wt)cDNA*, pG-*gcy-7p::clh-1(wt)cDNA*, pG-*dyf-11p::clh-1(wt)cDNA*, pG-*vap-1p::clh-1(wt) cDNA*, pG-*gcy-5p::SuperClomeleon* were created by site-specific recombination between pENTR and pDEST plasmids. Information of all plasmids used in this study can be found in *Supplementary file 3* (list and use) and *Supplementary file 4* (nucleotide sequences).

## Generation of transgenic animals

Expression constructs or genomic PCR fragments were injected at 0.1–50 ng/µl along with a co-injection marker (in many case pG-*myo-3p::venus* or pG-*lin-44p::GFP*, 5–10 ng/µl) and pPD49.26 (a gift from Andrew Fire, up to 100 ng/µl) as a carrier DNA. For comparison among genotypes, the transgene was initially introduced into wild-type background and transferred to other genetic backgrounds by cross. JN2249 was generated by injecting pG-*clh-1p::nls4::mTFP* and *lin-44p::mCherry* into an expression marker strain *Is[rimb-1p::nls4::mCherry; eat-4p::nls4::tagRFP; lin-44p::GFP]*.

## Fluorescence microscopy and measurement of sensory cilium length

Day one adults were mounted on 5% agar and anesthetized by 10 mM $NaN_3$, or 100 µM levamisole. Images were captured using a Leica HCX PL APO 40×/0.85 CORR CS objective or an HC PL APO 63×/0.40 CS objective on a Leica TCS-SP5 confocal microscope. The length of ASER sensory cilium was measured by simple neurite tracer plugin of ImageJ. All depicted representative fluorescence images were Z-stacked.

## Electrophysiology

We synthesized capped CLH-1 cRNAs using T7 mMESSAGE mMACHINE kit (Ambion) and purified by lithium chloride precipitation. cRNA was quantified by spectroscopy. Oocytes from *X. laevis* were prepared as follows. We anesthetized female *X. laevis* in cold MS-222 solution, and excised lobes of ovaries from a small incision made in the posterior ventral side. Oocytes were obtained by defolliculation; incubation of ovaries in 0.2% collagenase (Wako) in modified Barth's solution (MBSH; 88 mM NaCl, 1 mM KCl, 2.4 mM $NaHCO_3$, 0.82 mM $MgSO_4$, 0.33 mM $Ca(NO_3)_2$, 0.41 mM $CaCl_2$, 10 mM *N*-(2-hydroxyethyl)piperazine-*N*′−2-ethanesulfonic acid (HEPES, pH 7.6)) for 2 to 5 hr. Oocytes were washed and incubated at 16°C in MBSH with 100 U/mL penicillin and 0.1 mg/mL streptomycin overnight. Thereafter, oocytes were injected with cRNA for a final amount of 50 ng/oocyte. Oocytes were incubated in MBSH at 16°C for 2 days before recording. Currents were measured using a two-electrode voltage-clamp amplifier Oocyte Clamp OC-725C (Warner) at ambient temperature (22°C). Electrodes (0.3–1 MΩ) were filled with 3 M KCl, and then oocytes were perfused in an extracellular solution buffer with the following composition (in mM): 100 NaCl, 2 KCl, 1 $CaCl_2$, 2 $MgCl_2$, 10 HEPES, pH 7.2 for a standard recording condition (pH 7, Cl⁻). For extracellular acidification (pH 5, Cl⁻), 10 mM of 2-(*N*-morpholino)ethanesulfonic acid (pH 5.5) was used instead of 10 mM HEPES. To observe permeability of $HCO_3^-$ (pH 7, $HCO_3^-$), 85 mM NaCl was replaced with an equimolar amount

of $NaHCO_3$ (*Grant et al., 2015*). For data acquisition and analysis, pCLAMP suite of programs (Molecular Devices) were used.

## In vivo calcium imaging and chloride imaging

Ratiometric fluorescence reporters Yellow Cameleon 2.6 and SuperClomeleon were used for calcium and chloride imaging, respectively. We found no obvious defects in salt chemotaxis of the animals that carried these reporters in wild-type background. Imaging experiments were performed as described (*Iwata et al., 2011*; *Kunitomo et al., 2013*) with minor modifications. Worms were cultivated on standard NGM seeded with OP50 until adulthood and further incubated for 6 hr at distinct salt concentrations with or without food. Worms were then trapped in a polydimethylsiloxane (PDMS) microfluidics device (*Chronis et al., 2007*), and NaCl concentration steps were delivered to the animals' nose tip by switching imaging solutions (25 mM potassium phosphate (pH 6.0), 1 mM $CaCl_2$, 1 mM $MgSO_4$, 0.02% gelatin, NaCl at the indicated concentrations and glycerol to adjust their osmolarity to 350 mOsm). Time-lapse imaging was conducted with a DMI 6000B microscope (Leica) equipped with an HCX PL APO 63x objective (NA, 1.40), L5 filter set (a combination of 430/40 nm band-path excitation filter and a 40% transmittance ND filter at 535/40 nm dichromatic mirror, Leica), DualView (Filter sets: 505dcxr with 480/40 nm and 535/40 nm emission filters), and ImagEM EM-CCD camera (Hamamatsu) at two frames per second. All recordings were started 3 min after mounting to stabilize the light response of animals. Fluorescence intensity of the soma was measured. The region of interest (ROI) was tracked by Track Objects function of Metamorph software (Molecular Devices). For each frame, fluorescence intensity of the ROI was calculated by subtracting the background intensity (averaged fluorescence intensity adjacent to the ROI) from the average intensity of the ROI. The ratio of YFP/CFP fluorescence was referred to as $R$. $R_0$ was defined as the average of $R$ over 50 frames (25 s) prior to stimulation, and the fluorescence intensity ratio relative to $R_0$ ($R/R_0$) were calculated for a series of images. For traces, $R/R_0$ was averaged for all worms at each time point. The average $R/R_0$ change of SuperClomeleon was calculated as the difference between the value of last 10 s during 25 mM salt stimulus and that just prior to salt concentration change (*Figure 4f*; [averaged $R/R_0$ during B] - [averaged $R/R_0$ during A]). The average $R/R_0$ change of YC2.6 (e.g. *Figure 5b*) was calculated as the difference between the highest value of 10 s moving average of $R/R0$ during 25 mM NaCl stimulus ($R/R_0$ peak) and 10 s average of $R/R_0$ just prior to corresponding NaCl decrease (e.g. [averaged $R/R_0$ peak of B] - [averaged $R/R_0$ during A] for the 1st, [averaged $R/R_0$ peak of E] - [averaged $R/R_0$ during D] for the 2nd, [averaged $R/R_0$ peak of H] - [averaged $R/R_0$ during G] for the 3rd in *Figure 5a*). Note that the time point used for $R/R_0$ peak differs between individuals because rise speed was different between *clh-1* genotypes. $R/R_0$ decay1 (e.g. *Figure 5—figure supplement 1a*) was calculated as the difference between the $R/R_0$ peak and the average of last 10 sec $R/R_0$ during 25 mM NaCl stimulus (therefore just prior to 50 mM NaCl upstep; e.g. [averaged $R/R_0$ during B] - [averaged $R/R_0$ peak of C] for the 1 st, [averaged $R/R_0$ during E] - [averaged $R/R_0$ peak of F] for the 2nd, [averaged $R/R_0$ during H] - [averaged $R/R_0$ peak of I] for the 3rd, in *Figure 5a*). $R/R_0$ decay2 (e.g. *Figure 5—figure supplement 1b*) was calculated as the difference between the average of last 10 sec $R/R_0$ during 25 mM NaCl stimulus and the average of last 10 sec $R/R_0$ during the following 50 mM NaCl period (e.g. [averaged $R/R_0$ during C] - [averaged $R/R_0$ during D] for the 1 st, [averaged $R/R_0$ during F] - [averaged $R/R_0$ during G] for the 2nd, in *Figure 5a*).

## Quantification of the expression level of the *clh-1* promoter

Worms that carry *clh-1p::nls4::mTFP* and *gcy-5p::nls4::mCherry* were cultivated on standard NGM seeded with OP50 until adulthood. One-day adults were further incubated at 0 mM or 100 mM NaCl with food for 6 hr if needed. Naive (directly prepared from the cultivation plate) or different salt concentration-experienced animals were mounted on 5% agar and anesthetized by 10 mM $NaN_3$. Images were captured using a Leica HC PL APO 10×/0.40 CS objective on a Leica TCS-SP5 confocal microscope. The expression level of *clh-1* in ASER of each animal was determined as described previously with modifications (*Nagashima et al., 2019*). The fluorescence intensity ratio of mTFP to mCherry (mTFP/mCherry) from the confocal image that gave largest area of ASER nucleus was defined as *clh-1* expression level.

## Quantitative analysis of animals' behavior

Quantitative behavior analysis was conducted as described previously with modifications (*Jang et al., 2019*; *Kunitomo et al., 2013*). Briefly, animals and chemotaxis assay plates were prepared as described above in salt chemotaxis assay except that $NaN_3$ was omitted. To reduce the chance of collision of worms, only 30–50 worms were placed. Images of whole assay plate were acquired for 15 min at one frame per second. Probability of pirouette and curving rate were calculated as previously described (*Jang et al., 2019*). Pirouette index was defined as the difference of pirouette probability between negative $dC/dt$ rank and positive $dC/dt$ rank. Weathervane index was defined as the slope of the regression line in relationship between NaCl gradient in normal direction and the curving rate. We exploited data in the range of −0.3 to 0.3 for $dC/dt$ and −2 to 2 for $dC/dn$ because both pirouette probability and curving rate converged toward zero with high variability in the range of highly negative or positive $dC/dt$ or $dC/dn$ range, probably due to small number of data points at the gradient peaks. We calculated data of each plate, then showed average and s. e.m. in figures. Data from 40 to 540 s were used to calculate the behavioral parameters because trajectories of worms were highly interrupted by collision of worms before 40 s.

## Calcium imaging of AIB in freely moving animal

Animals expressed GCaMP6s and mCherry in AIB (JN3329, see *Supplementary file 1*). Worms were cultivated on standard NGM until young adulthood and further incubated overnight on NGM plates with 0 mM NaCl. A total of 20 ~ 25 worms were washed out from the plate, and injected into a PDMS microfluidic device (*Albrecht and Bargmann, 2011*). After few minutes of acclimation to the PDMS environment under continuous flow of imaging solution without NaCl (see In vivo calcium imaging and chloride imaging for composition), a 25 mM NaCl up-step was delivered to worms. Bright-field images for locomotion analysis were acquired at 33 frames per second and fluorescence images for AIB activity were acquired at 4 frames per second with a BX51 microscope (Olympus) equipped with a halogen light source (U-LH100IR), a motorized stage (HV-STU02- 1, HawkVision), an LMPlanF1 5x objective (NA, 0.13), U-25ND25 (Olympus), a combination of 480/40 nm band-path excitation filter and a 25% transmittance ND filter at 505/40 nm dichromatic mirror (Leica), DualView (Filter sets: 565dcxr with 520/30 nm and 630/50 nm emission filters), and a CCD camera (GRAS-c3K2M-C, Point Grey Research). Tracking of particular animal was performed using a Linux-based software (*Satoh et al., 2014*). After subtracting background, fluorescence intensity of GCaMP6s was normalized by that of mCherry. Average fluorescence intensity over 50 frames (12.5 s) prior to stimulation was set as $F_0$ and the fluorescence intensity relative to $F_0$ ($F/F_0$) were calculated for a series of images. For traces, $F/F_0$ was averaged for all worms at each time point. The $F/F_0$ change in response to stimulation was calculated as the difference of averaged $F/F_0$ between 125 to 175 frames for peak and 51 to 100 frames for baseline (i.e. [averaged $F/F_0$ during B] - [averaged $F/F_0$ during A] in *Figure 6e*). Reversal was defined as backward movement whose velocity was less than 0 µm/s (*Figure 6—figure supplement 2*, colored in red). In *Figure 6d*, animals those showed obvious reversal response (reversed longer than 35 frames (1 s) within 501 frames (15 s) after salt up-step stimulus at velocity less than −100 µm/s) were counted as 'reversed animals' to exclude short spontaneous reversals that occur independent of salt stimulus (*Figure 6—figure supplement 2*, short red bouts).

## Data analyses

The sample sizes were experimentally determined, with referring to those previously reported. Repeats of most experiments were performed in 3–6 separate days. Statistical analyses were performed using Prism v.5 (GraphPad software, San Diego, CA).

## Acknowledgements

We thank the Caenorhabditis Genetics Center (CGC) and the National BioResource Project (NBRP) for strains; Ishihara T for pDEST-*nls-YC2.60*, Oka Y for electrophysiology setup and technical assistance, Chronis N, Albrecht D and Bargmann C for design of microfluidics chips. We also thank members of the Iino laboratory members for helpful comments and discussion.

# Additional information

## Competing interests
Yuichi Iino: Reviewing editor, *eLife*. The other authors declare that no competing interests exist.

## Funding

| Funder | Grant reference number | Author |
| --- | --- | --- |
| Japan Society for the Promotion of Science | Grant-in-Aid for Scientific Research (S) JP17H06113 | Yuichi Iino |
| Japan Science and Technology Agency | CREST JP17H06113 | Yuichi Iino |
| Japan Society for the Promotion of Science | Grants-in-Aid for Innovative Area "Artificial Intelligence and Brain Science" 19H04980 | Yuichi Iino |
| University of Tokyo | Center for Integrative Science of Human Behavior (CiSHuB) | Yuichi Iino |
| Japan Society for the Promotion of Science | Grant-in-Aid for Scientific Research | Hirofumi Sato |
| Japan Society for the Promotion of Science | Grants-in-Aid for Innovative Area 18H04881 | Shinji Kanda |
| Japan Society for the Promotion of Science | Grant-in-Aid for challenging Exploratory Research 18K19323 | Shinji Kanda |
| Japan Society for the Promotion of Science | Grant-in-Aid for Scientific Research 19K06952 | Hirofumi Kunitomo |
| Salt Science Research Foundation | No. 1728 and No. 2043 | Hirofumi Kunitomo |
| Japan Society for the Promotion of Science | Grant-in-Aid for Young Scientists 19K1628 | Hirofumi Sato |

The funders had no role in study design, data collection and interpretation, or the decision to submit the work for publication.

## Author contributions
Chanhyun Park, Resources, Formal analysis, Validation, Investigation, Visualization, Methodology, Writing - original draft, Writing - review and editing; Yuki Sakurai, Resources, Validation, Investigation; Hirofumi Sato, Resources, Investigation, Methodology; Shinji Kanda, Resources, Supervision, Funding acquisition, Writing - review and editing; Yuichi Iino, Conceptualization, Formal analysis, Supervision, Funding acquisition, Writing - review and editing; Hirofumi Kunitomo, Conceptualization, Resources, Supervision, Funding acquisition, Validation, Investigation, Writing - original draft, Writing - review and editing

## Author ORCIDs
Chanhyun Park  https://orcid.org/0000-0002-0200-6903
Yuichi Iino  http://orcid.org/0000-0002-0936-2660
Hirofumi Kunitomo  https://orcid.org/0000-0001-7312-7051

## Decision letter and Author response
Decision letter https://doi.org/10.7554/eLife.55701.sa1
Author response https://doi.org/10.7554/eLife.55701.sa2

# Additional files

## Supplementary files
- Supplementary file 1. Strain list.
- Supplementary file 2. Genotyping primer.
- Supplementary file 3. Plasmid list and use.
- Supplementary file 4. Plasmid sequences.
- Supplementary file 5. Statistics.
- Transparent reporting form

## Data availability

All data analysed during this study are included in the manuscript and supporting files. Detailed statistical reporting appears in supplementary file 5. Other source data files have been uploaded to Dryad Digital Repository (https://doi.org/10.5061/dryad.4tmpg4f8c).

The following dataset was generated:

| Author(s) | Year | Dataset title | Dataset URL | Database and Identifier |
|---|---|---|---|---|
| Park C, Sakurai Y, Sato H, Kanda S, Iino Y, Kunitomo H | 2020 | Roles of the ClC chloride channel CLH-1 in food-associated salt chemotaxis behavior of C. elegans | https://dx.doi.org/10.5061/dryad.4tmpg4f8c | Dryad Digital Repository, 10.5061/dryad.4tmpg4f8c |

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
