## [Decision Letter]

**Acceptance summary:**

In this work, the authors describe the role of disrupted chloride ion homeostasis in regulating the responses of the *C. elegans* ASER sensory neurons to repeated salt stimuli. The authors find that altered salt-evoked chloride dynamics in ASER decrease the ability of *C. elegans* to navigate salt gradients and migrate towards low salt concentrations associated with cultivation in the presence of bacterial food.

**Decision letter after peer review:**

Thank you for submitting your article "Roles of the ClC chloride channel CLH-1 in food-associated salt chemotaxis behavior of *C. elegans*" for consideration by *eLife*. Your article has been reviewed by two peer reviewers, and the evaluation has been overseen by a Reviewing Editor and Richard Aldrich as the Senior Editor. The reviewers have opted to remain anonymous.

The reviewers have discussed the reviews with one another and the Reviewing Editor has drafted this decision to help you prepare a revised submission. Please aim to submit the revised version within two months.

Summary:

Both reviewers agreed that the work describing the contributions of chloride channels to salt chemotaxis is interesting. In particular, the identification of the neomorphic alleles is of interest. However, reviewers agreed that a few points need to be addressed experimentally to support the conclusions. We think that these points should be readily addressable within 2 months (please ask for extensions if needed based on the current crisis).

Essential revisions:

1) Additional characterization/description of the two new *clh-1* alleles:

a) Explain textually or via experiments: The authors suggest that the pe mutants may increase Cl^-^ flux, however, data shown in Figure 4a-e show the opposite. Currents are actually smaller and currents shown in panel C are particularly small and seem to be less rectifying. This is also seen in the IV plot.

b) Is there any literature in which similar mutations in other ClC channels have been studied? If so, please reference them.

c) Characterize important channel properties such as sensitivity to pH and permeability properties to determine how those are affected by the mutations.

2) Figure 2. These are interesting results. However, to test the possibility that *clh-1(pe)* mutations impair other CLH proteins, the authors should have addressed this question using the *clh-1(pe)* mutant worms, especially since they have already established that the *clh-1* KO animals have normal salt chemotaxis. The authors should create double mutants with the *clh-1(pe)* strains.

3) Consider the possibility that expression changes of ClC channels in WT and *clh-1(pe)* mutants in different salt conditions on the plate may account for some of the observed phenotypes (Figure 5 and supplements).

4) AIB calcium imaging results in Figure 6: The defect in the pe alleles are consistent with the behavioral defects, but the effect appears to be mild. It will be informative to test whether the defects in AIB are generated by the malfunction of *clh-1(pe)* in ASER.

[Editors' note: further revisions were suggested prior to acceptance, as described below.]

Thank you for submitting your article "Roles of the ClC chloride channel CLH-1 in food-associated salt chemotaxis behavior of *C. elegans*" for consideration by *eLife*. Your article has been reviewed by one peer reviewers, and the evaluation has been overseen by a Reviewing Editor and Richard Aldrich as the Senior Editor. The reviewer has opted to remain anonymous.

We are pleased to say that the paper is in principle, appropriate for publication in *eLife*. However, prior to final acceptance, please revise the manuscript for clarity and presentation as indicated in the comments below.

*C. elegans* navigates salt gradients in an experience-dependent manner. Here the authors report the contribution of chloride channels in modulating neuronal excitability of the salt-sensing ASER sensory neuron, and show that these channels are required for the animal to correctly perform chemotaxis on salt gradients based on feeding experience.

1) Figure 1—figure supplement 1c and Figure 2—figure supplement 1c: The scale of Y axis is too big for the data represented. Authors should use a break in the Y axis so data can be compared more easily.

2) Figure 5 with all its supplemental figures and Figure 6: the bar graphs do not show individual data points. The authors should add individual data points to all the bar graphs across all their figures.

3) The authors show both chloride and calcium measurements. They should clearly indicate in the panels whether the data shown indicate Cl^-^ or Ca^2+^ measurements so that readers can more easily follow the story. Alternatively, they could use different colors in the graphs, again to make the story easier to follow.

4) Results third paragraph: Do the authors here mean Figure 1—figure supplement 1d and e?

5) Subsection “ASER salt response is altered in *clh-1* mutants”, the authors state that there is no difference between the different genotypes in the expression of *clh-1* transcriptional reporter (Figure 5—figure supplement 1e). But actually in wt cultivated in 0 mM there is a significant reduction. This is expected to cause reduced Cl^-^ influx in wt under these conditions. This phenomenon is not seen in the pe mutants. So in the pe mutants there should be more Cl^-^ influx which is consistent with what the authors see. The authors should revise this paragraph to reflex this observation.

6) The authors should also either quantify data shown in Figure 3D or state that it is not known whether the pe variants are expressed at higher or lower levels than wild type.

---

## [Author Response]

Essential revisions:1) Additional characterization/description of the two new *clh-1* alleles:

We appreciate the reviewers for thoughtful considerations and valuable suggestions. In the revised manuscript, we replicated and extended electrophysiological experiments using *Xenopus* oocytes to verify the previous results and to further characterize the properties of CLH-1*(pe)* mutants. In summary, we did not find remarkable difference between CLH-1*(wt)* and CLH-1*(pe)* mutants except that averaged currents were smaller in the mutants. CLH-1*(pe)* mutants retained pH sensitivity. According to these results and considering the possibility that SuperClomeleon is sensitive to pH change, we modified the model and discussed possible reasons for disagreement between chloride imaging and electrophysiology (Figure 7 and the Discussion section). Details are described below.

a) Explain textually or via experiments: The authors suggest that the *pe* mutants may increase Cl^-^ flux, however, data shown in Figure 4a-e show the opposite. Currents are actually smaller and currents shown in panel C are particularly small and seem to be less rectifying. This is also seen in the IV plot.

In revision, we initially replicated measurement of membrane currents in the oocytes that expressed either CLH-1*(wt),* CLH-1(*pe572)* or CLH-1*(pe577)* using freshly prepared materials under the standard bath condition (pH 7.2 and 100 mM NaCl, details are described in Materials and methods). The currents from individual oocytes that expressed CLH-1*(pe)* mutants were largely overlapped with that of CLH-1*(wt)* (Figure 4—figure supplement 2a-c, pH 7), whereas averaged currents were actually smaller in the mutants (Figure 4e and Figure 4—figure supplement 1), which was in contrast with the result of chloride imaging of ASER.

However, interpretation of the whole-cell membrane currents using *Xenopus* oocytes requires careful consideration (reviewed in Bianchi and Driscoll, 2006; Dascal, 2000). First, cellular environment of *Xenopus* oocyte differs from that of *C. elegans* neurons. Inward currents were observed in membrane potential range as low as -80 mV or lower in electrophysiology, whereas chloride imaging was performed under the condition in which ASER was rather depolarized; a recent report showed that ASER had resting membrane potentials of approximately −60 mV and a salt down-step stimulus induced depolarization (Shindou et al., 2019). In addition, the CBS domains are known to regulate the activity of ClC channels through phosphorylation or binding to auxiliary subunits (Dave et al., 2010; Denton et al., 2005; Estévez et al., 2004; Stölting et al., 2013). Given that CLH-1 also contains the CBS domains, such regulations may be required for intrinsic activity of CLH-1, although it needs to be experimentally validated. Second, the expression level of channel proteins may affect whole-cell current. Although we adjusted the amount and quality of cRNA injected into oocytes, average expression level of the CLH-1*(pe)* mutants might be lower than that of wild type; efficiency of membrane transport or stability of channel proteins might be affected by the mutations. Future studies such as single-channel analyses are required to examine these possibilities.

Nonetheless, we considered the possibility that reduction of CLH-1 activity may enhance chloride influx into ASER upon depolarization. The chloride potential of plasma membrane is regulated by the balance of transporters and channels activities. If intrusion of chloride into ASER was mainly mediated by CLH-1 in the resting state, reduced CLH-1 activity in the mutants would result in a stronger driving force for intrusion of chloride into the cell compared to wild type. We modified the model accordingly (Figure 7, middle right) and added these discussions in the text.

In the replicated experiments, CLH-1-dependent currents were obvious only at hyperpolarized membrane potentials in both wild type and mutants (Figure 4—figure supplement 1j-l). We did not pursue the rectification properties further.

b) Is there any literature in which similar mutations in other ClC channels have been studied? If so, please reference them.

A substantial number of mutations have been characterized so far in ClC genes in human genetic diseases and animal models. In addition, extensive structure-function analyses have identified residues that are involved in gating and ion permeability of ClC channels/transporters, most of which were studied in mammalian proteins (reviewed in Jentsch and Pusch, 2018). Many of these variations resulted in reduction- or loss-of-function, while some hypermorphic mutations that change permeability and gating property of ClC-Kb and ClC-2 have been reported (Fernandes-Rosa et al., 2018; Frey et al., 2006; Göppner et al., 2019; Schewe et al., 2019; Scholl et al., 2018; Stölting et al., 2014). To our knowledge, none of these mutations was similar to *clh-1(pe)* mutations.

*clh-3* is the best-characterized ClC gene in *C. elegans*. It encodes two splicing variants, CLH3a and CLH-3b, and regulates the activity of the hermaphrodite-specific neurons (HSNs) that control egg-laying behavior. A gain-of-function mutation of *clh-3* with an increased channel activity inhibited HSN by decreasing its excitability (Branicky et al., 2014). The mutation was located in the subunit interface. CLH-3b is activated during meiotic maturation in oocytes and regulate ovulation (Rutledge et al., 2001). An intracellular loop (referred to as H-I loop) between two α helices H and I is involved in regulation of channel activity through interacting with the CBS2 domain (Feng et al., 2010; Yamada et al., 2013). Homology modeling predicts the Met293 residue of CLH-1 is equivalent of Ile219 in CLH-3b, and these residues are incorporated in a membrane helix adjacent to H-I loop (Figure 1—figure supplement 2b and Figure 2—figure supplement 1a). Substitution of Ile219 to phenylalanine in CLH-3b had no effect on channel activity (Yamada and Strange, 2016). It is interesting that isoleucine is the conserved residue in CLH genes at this position, but methionine in CLH-1, which was substituted to isoleucine by *clh-1(pe572)* mutation (Figure 2—figure supplement 1a). We added a sentence “To date, substantial number of mutations have been characterized in ClC genes, most of which are mammalian ClCs. To our knowledge, however, none of these mutations was similar to *clh-1(pe)* mutations.” in the Discussion section.

c) Characterize important channel properties such as sensitivity to pH and permeability properties to determine how those are affected by the mutations.

CLH-1 is activated by extracellular acidification and permeable to HCO_3_^-^ (Grant et al., 2015). In the revised manuscript, we measured whole-cell currents under the conditions with i) reduced extracellular pH to 5.5 and ii) 85 mM of extracellular chloride was replaced with an equimolar amount of HCO_3_^-^.

The currents were increased by extracellular acidification in CLH-1*(pe)* mutants as same as wild type (Figure 4—figure supplement 1 and Figure 4—figure supplement 2a-c), indicating that sensitivity to pH is retained in the mutants. It has also been reported that CLH-1 currents were comparable between the two conditions in which dominant extracellular anion was Cl^-^ and HCO_3_^-^(Grant et al., 2015). We obtained similar results with CLH-1(*wt*) as well as CLH-1(*pe*) mutants (Figure 4—figure supplement 1). We added these results in the revised manuscript.

To characterize the permeability of CLH-1(*wt*) and CLH-1(*pe*) mutants, we calculated the reversal potentials (Author response table 1). These results indicated a relative permeability of HCO_3_^-^ > Cl^-^ for the all CLH-1 variants. However, large deviation especially in HCO_3_^-^ experiments and unsuccessful recordings using I^-^ or Br^-^ ions in our experimental setup (not shown) precluded us to describe permeability properties of the CLH-1 variants. Replacement of 85 mM chloride to equimolar gluconate generated inward currents at hyperpolarized membrane potential in both CLH-1(*wt*) and CLH-1(*pe*) mutants as reported previously (Grant et al., 2015 and Author response image 1). These results show that under our experimental condition the properties of CLH-1 as a voltage-dependent, extracellular pH-sensitive anion channel were retained in CLH-1*(pe)* mutants.

**Author response table 1. resptable1:** The reversal potential of wild-type and mutant CLH-1 channels for Cl^-^ and HCO_3_^-^ at pH7. Number of samples in parentheses, unit: mV.

	Cl-	HCO3-
CLH-1(*wt*)	-20.89 ± 5.03 (20)	-26.07 ± 14.08 (19)
CLH-1(*pe572*)	-19.30 ± 4.44 (18)	-29.50 ± 16.36 (18)
CLH-1(*pe577*)	-14.20 ± 11.46 (8)	-19.69 ± 9.92 (8)

**Author response image 1. sa2fig1:** Whole cell currents of the *Xenopus oocytes* that express CLH-1 under the presence of extracellular gluconate.

2) Figure 2. These are interesting results. However, to test the possibility that *chl-1(pe)* mutations impair other CLH proteins, the authors should have addressed this question using the *clh-1(pe)* mutant worms, especially since they have already established that the *clh-1* KO animals have normal salt chemotaxis. The authors should create double mutants with the *clh-1(pe)* strains.

We thank the reviewer for the critical comment to improve the manuscript. We generated a series of double (or triple in the case of *clh1(pe572) clh-2 clh-3*) mutants that carry *clh-1(pe572)* and a deletion in another *clh* gene and examined their salt chemotaxis phenotype (Figure 2a). We found that all double mutants showed a salt chemotaxis defect similar to that of the *clh-1(pe572)* single mutant, that is, defective chemotaxis toward low salt. Furthermore, a hexatruple mutant that carry *clh-1(pe572)* and a deletion in five other *clh* genes also showed salt chemotaxis similar to that of *clh-1(pe572)* single mutant (Figure 2c). These results, with the results of single and multiple *clh* deletion mutants, indicate that salt chemotaxis defect of *clh-1(pe572)* mutant is not attributed to impairment of other CLH proteins. We added these results in the text.

Nevertheless, analyses using *clh-1(tm1243)* putative null allele suggested that *clh* genes may redundantly act in salt chemotaxis. We generated another hexatruple mutant that carry *clh-1(tm1243)*. This hexatruple mutant showed chemotaxis defects toward both high and low salt concentrations (Figure 2c), which had not been obvious in the quadruple mutants (Figure 2b). Chemotaxis defect to low salt was more severe in the *clh-1(pe572)* hexatruple mutant than the *clh-1(tm1243)* hexatruple mutant (Figure 2c). These results further suggest that the effect of *clh-1(pe572)* missense mutation is not simply caused by inhibition of other CLH proteins.

3) Consider the possibility that expression changes of ClC channels in WT and *clh-1(pe)* mutants in different salt conditions on the plate may account for some of the observed phenotypes (Figure 5 and supplements).

Our genetic analyses indicated that the *clh-1(pe)* mutations are neomorphic or possibly hypermorphic (Figure 1c, Figure 1—figure supplement 3c and d, and Figure 2), and the mutations affected salt chemotaxis in dose-dependent manner (Figure 1—figure supplement 3b). In addition, the defects were obvious after cultivation at 0 mM NaCl rather than 100 mM (Figure 1b, Figure 5 and supplements). These results might imply a possibility that *clh-1* expression changed (increased) under low NaCl concentrations. In the revised manuscript, we examined the expression level of *clh-1* by quantifying the fluorescence intensity of a *clh-1p::mTFP* reporter in ASER (Figure 5—figure supplement 1e). We noticed an increased expression level in *clh-1(pe572)* background. However, considering that *pe572* and *pe577* result in similar phenotype and overall expression level of *clh-1(pe577)* was similar to that of wild type, we concluded that expression change of *clh-1* was not the major reason for the phenotype of *clh-1(pe)* mutants. We added these explanations in the text.

4) AIB calcium imaging results in Figure 6: The defect in the pe alleles are consistent with the behavioral defects, but the effect appears to be mild. It will be informative to test whether the defects in AIB are generated by the malfunction of *clh-1(pe)* in ASER.

We appreciate the reviewer for valuable suggestions. Importance of ASER was shown in salt chemotaxis phenotype (Figure 3b and c, Figure 3—figure supplement 1a). However, as the reviewer pointed out, it was not clear whether the defects of AIB response and the reduction of behavioral response were attributed to ASER abnormality. To address this, we observed phenotype of *clh-1(pe572)* mutants whose ASER was rescued by cell-specific expression of *clh-1(wt)*. We found that both calcium response of AIB and behavioral response of animals were restored in the rescued animals (Figure 6c-f, symbols colored in grey and Figure 6—figure supplement 2). Furthermore, in the revised manuscript, we examined whether expression of *clh-1(wt)* rescued salt responses of ASER. Cell-specific expression of *clh-1(wt)* cDNA in ASER restored calcium dynamics of ASER in response to salt stimuli (Figure 5a, b and Figure 5—figure supplement 1a, b, symbols colored in grey). Likewise, the response of SuperClomeleon was also restored (Figure 4f, g, symbols colored in grey), confirming that the defects of ASER were compensated by expression of CLH-1*(wt)* in ASER. These results strongly indicate that the aberrant responses of ASER to salt stimulation caused the defects of AIB and behavioral responses in *clh-1(pe572)* mutants. We described these results in the text.

[Editors' note: further revisions were suggested prior to acceptance, as described below.]

We are pleased to say that the paper is in principle, appropriate for publication in eLife. However, prior to final acceptance, please revise the manuscript for clarity and presentation as indicated in the comments below.*C. elegans* navigates salt gradients in an experience-dependent manner. Here the authors report the contribution of chloride channels in modulating neuronal excitability of the salt-sensing ASER sensory neuron, and show that these channels are required for the animal to correctly perform chemotaxis on salt gradients based on feeding experience.1) Figure 1—figure supplement 1c and Figure 2—figure supplement 1c: The scale of Y axis is too big for the data represented. Authors should use a break in the Y axis so data can be compared more easily.

We appreciate this suggestion to improve visibility of the data. We expanded the scale of Y axis at the range of bar graphs and inserted a break to indicate the maximum of the scale (that is, 1) so that the values of immobility index can be easily compared not only within the same panel but also between different figures.

2) Figure 5 with all its supplemental figures and Figure 6: the bar graphs do not show individual data points. The authors should add individual data points to all the bar graphs across all their figures.

According to the reviewer’s suggestion, we presented individual data points to all bar graphs. This might have made the means and error bars somewhat difficult to see in some cases of Figure 5 with its supplemental figures and Figure 6 due to the substantial number of points (typically, more than 10). We added individual data points also in Figure 3—Figure supplement 1d, e, and Figure 4g in the revised manuscript.

3) The authors show both chloride and calcium measurements. They should clearly indicate in the panels whether the data shown indicate Cl^-^ or Ca^2+^ measurements so that readers can more easily follow the story. Alternatively, they could use different colors in the graphs, again to make the story easier to follow.

We appreciate this suggestion. In the revised manuscript, we indicated the name of cell observed and ion species measured in addition to animal’s cultivation condition in all panels that show traces of chloride or calcium indicators. For example, for Figure 4f, “0 mM fed, ASER, Cl^-^ measurements”.

4) Results third paragraph: Do the authors here mean Figure 1—figure supplement 1d and e?

We thank the reviewer for this comment. In the original manuscript, we wanted to refer the gene structure of the deletion mutations (Figure 1—figure supplement 2a) and salt chemotaxis phenotype of the mutants (Figure 1—figure supplement 3a). As the reviewer pointed out, however, Figure 1—figure supplement 1d and 1e also demonstrate salt chemotaxis phenotype of the *clh1(tm1243)* deletion mutants. We revised the sentence as follows:

“Interestingly, deletion mutants of *clh-1*, all of which harbor a lesion in the pore-forming transmembrane domain of CLH-1 and hence are putative loss-of-function alleles (Figure 1—figure supplement 2a), showed almost no discernible defect in salt chemotaxis (Figure 1—figure supplement 3a and Figure 1—figure supplement 1d, e).”.

5) Subsection “ASER salt response is altered in *clh-1* mutants”, the authors state that there is no difference between the different genotypes in the expression of *clh-1* transcriptional reporter (Figure 5—figure supplement 1e). But actually in wt cultivated in 0 mM there is a significant reduction. This is expected to cause reduced Cl^-^ influx in wt under these conditions. This phenomenon is not seen in the pe mutants. So in the pe mutants there should be more Cl^-^ influx which is consistent with what the authors see. The authors should revise this paragraph to reflex this observation.

We appreciate the reviewer for this insightful suggestion. As the reviewer pointed out, reduction of *clh-1* expression in ASER in wild type, which was not observed in *clh-1(pe)* mutants, may be responsible for the small Cl^-^ influx upon salt stimulus after cultivation at 0 mM NaCl. This may further account for the difference of salt stimulus-induced Ca^2+^ responses between wild type and the mutants. We revised the sentence to precisely describe the results and discussed the abovementioned possibility as follows:

“We observed expression of a *clh-1* promoter-driven transcriptional reporter in ASER and found a reduction of its expression in wild type, but not in the *clh-1(pe)* mutants, after cultivation at 0 mM NaCl (Figure 5—figure supplement 1e). This result raises a possibility that the expression change of CLH-1 in ASER might contribute to the difference of chloride and calcium responses of the cell between wild type and the *clh-1(pe)* mutants”.

6) The authors should also either quantify data shown in Figure 3d or state that it is not known whether the pe variants are expressed at higher or lower levels than wild type.

According to the reviewer’s suggestion, we added a following sentence. “It is unknown whether the CLH-1*(pe)* variants are expressed at higher or lower levels than CLH-1*(wt)*, although expression level of a *clh-1* promoter-driven reporter slightly differed among genotypes (Figure 5—figure supplement 1e)”

We did not quantify CLH-1::mTFP in Figure 3d, because the reporter was driven by an ASER specific *gcy-5* promoter to observe subcellular localization of CLH-1, not by the *clh-1* promoter that would reflect authentic expression pattern of the gene.